# Adverse health correlates of intimate partner violence against older women: Mining electronic health records

Serhan Yılmaz[1], Erkan Gunay[2], Da Hee Lee[3], Kathleen Whiting[4], Kristin Silver[5], Mehmet Koyuturk[1,6], Gunnur Karakurt[7]*

1 Department of Computer and Data Sciences, Case Western Reserve University, Cleveland, OH, United States of America, 2 Emergency Department, Şişli Hamidiye Etfal Training and Research Hospital, Istanbul, Turkey, 3 Osteopathic Medicine and Public Health, Des Moines University, Des Moines, IA, United States of America, 4 Neuroscience Program, Uniformed Services University, Washington, DC, United States of America, 5 Behavioral Health, Center of Outpatient Education, VA Northeast Ohio Healthcare System, Cleveland, OH, United States of America, 6 Center for Proteomics & Bioinformatics, Case Western Reserve University, Cleveland, OH, United States of America, 7 Department of Psychiatry, Case Western Reserve University, Cleveland, OH, United States of America

* gkk6@case.edu

**Data Availability Statement:** The IBM Explorys Therapeutic Dataset are run by IBM who makes the data available for secondary use (for example, scientific research) on a commercial basis. Data were used under license for the study presented in

## Abstract

Intimate partner violence (IPV) is often studied as a problem that predominantly affects younger women. However, studies show that older women are also frequently victims of abuse even though the physical effects of abuse are harder to detect. In this study, we mined the electronic health records (EHR) available through IBM Explorys to identify health correlates of IPV that are specific to older women. Our analyses suggested that diagnostic terms that are co-morbid with IPV in older women are dominated by substance abuse and associated toxicities. When we considered differential co-morbidity, i.e., terms that are significantly more associated with IPV in older women compared to younger women, we identified terms spanning mental health issues, musculoskeletal issues, neoplasms, and disorders of various organ systems including skin, ears, nose and throat. Our findings provide pointers for further investigation in understanding the health effects of IPV among older women, as well as potential markers that can be used for screening IPV.

## Introduction

Intimate Partner Violence (IPV) is a devastating public health problem and affects millions of women globally each year. According to recent statistics, about a quarter of women in the US experience severe physical violence from their partner in their lifetime [1]. This rate varies from 15 percent to 71 percent around the world [2]. IPV can be broadly defined as "abusive behaviors perpetrated by someone who is or was involved in an intimate relationship with the victim" [1]. IPV involves physical, emotional, sexual harm to the victim-survivor [1, 3]. Other common forms of abuse include psychological/emotional maltreatment through behaviors that causes emotional pain or injury with verbal threats, berating, harassment, or intimidation,

this manuscript in the form of aggregate statistics (number of records) in a specified population (individual records were never used by the study because of patient agreements to protect the privacy of the patients). In Supplementary Data 1, we provide the raw data obtained from IBM Explorys (i.e., number of records for the terms in the revelant populations). In Supplementary Data 2 and 3, we provide the terms identified in our analyses and their estimated co-morbidity levels for intimate partner violence.

**Funding:** Gunnur Karakurt (PI) This publication was made possible by R01-LM012518 from the National Library of Medicine. Its contents are solely the responsibility of the authors and do not necessarily represent the official views of the NIH. The funders had no role in study design, data collection and analysis, decision to publish, or preparation of the manuscript.

**Competing interests:** The authors have declared that no competing interests exist.

economic deception, and willful negligence [4–6]. The reported adverse health effects of IPV extend from minor injuries and cuts, chronic conditions to acute severe injuries, and even death [7–10]. Past research additionally indicated that mental health-related issues such as depression, anxiety, post-traumatic stress disorder (PTSD), substance abuse, and suicide are widely observed among survivors of IPV [11, 12].

IPV has been studied mainly as a problem that predominantly affects younger women. The U.S. Department of Health and Human Services recommends IPV screening for women and girls ages 15–46. While much evidence exists documenting the most severe forms of relationship violence that are directed against women of childbearing age, older women are also vulnerable to IPV at an increasing rate [7, 13, 14]. Older adult women report that nonphysical abuse can also be harmful to the victim's mental and physical health [15]. Neglect, defined as the failure of a caregiver to fulfill his/her duties, including behaviors such as withholding food or medication, also affects older women and can also have detrimental health effects [4, 6].

It can be challenging to detect the physical effects of IPV among older women, since they are naturally more prone to injury and ailment [8, 16]. Furthermore, as health declines over the years, health care providers may mistake the signs of abuse as normal wear and tear to physical and mental health. Limited research on the health of older female victims of partner violence shows that health problems reported by older women are concordant with the general population [10, 17]. Older women who report nonphysical abuse such as seclusion or exclusion, financial exploitation also report that these forms of abuse adversely affects their well being [5, 15, 18]. However, there is limited information on the specific health consequences of IPV on aging women [19–21]. In this paper, we aim to identify health correlates of IPV that are specifically common in older adult women.

A complicating factor for researchers and service professionals is the lack of coordination between the fields of IPV and older adult abuse. The lack of conceptual clarity on older women abuse intersecting with IPV presents many challenges to understanding victims' experiences and providing necessary support [22]. Barriers to diagnosis and treatment include the victim's fear of reprisal by the abuser, victim denial or shame, inexperience and lack of knowledge by health care personnel, and the ageist attitude of society [16]. Furthermore, older adult women who are being abused may not be as familiar with the language or concepts used to describe violence and may not have the willingness or ability to disclose such events [23]. Finally, the victim may feel too ashamed to admit the abuse is occurring or may be frightened by the prospect of living alone after many years of co-dependence [23]. All these barriers make this population harder to reach and may prevent the victim from asking for help or making any progress toward leaving their abuser. For these reasons, identification of potential health-related markers of IPV in older women can also be useful for clinicians, care providers, and service professional to identify potential signs of IPV and develop strategies to follow up accordingly.

We take a data-driven approach to identify the health correlates of IPV against older women. Specifically, we aim to answer the following questions:

1. What are the conditions that are observed commonly in women (particularly older women) who suffer from IPV?

2. Which of these conditions are more frequently observed in older women as compared to younger women?

To answer these questions, we utilize electronic health records (EHRs) provided by the IBM Explorys Therapeutic Dataset [24]. IBM Explorys is a private Electronic Health Record (EHR) database, which pools data from more than 8 billion ambulatory visits to more than 40

US healthcare networks including diverse institutions and points of care [25]. It is a browser-based search engine with query options of various diagnostic categories based on ICD-9/10 codes. Cohorts include data on diagnoses, findings, and demographics. In this paper, we use diagnostic data we obtain by querying this tool. Throughout this paper, we refer to diagnoses, findings, and demographics returned by Explorys as "terms".

Records for patients 18 years or older seen in multiple healthcare systems from 1999 to 2019 are included in the database. Data are standardized and normalized using common ontologies, searchable through a HIPAA-compliant, patient de-identified web application (Explore; Explorys Inc). The diversity of pooled data in IBM Explorys is aimed at reflecting the full real-world healthcare continuum, while the large patient cohort enhances statistical power. Moreover, it allows flexible queries to acquire data that represent a specific population (such as older women who suffer from IPV). To identify records that belong to older women, we query the database for women age 65 and over. Accordingly, we use the term "older adult" through-out the paper to indicate adults with the chronological age 65 and over.

While the richness of data and the flexibility of queries in IBM Explorys provide unprece-dented opportunities for mining data to identify previously unreported associations, there are important computational and statistical challenges due to the employed privacy measures: (i) IBM Explorys does not provide access to individual records and allows querying of the rec-ords only in the form of number of records, and (ii) the number of records provided in query results are rounded to the nearest ten, posing further challenges to assess statistical significance because of the additional uncertainty due to. For these reasons, it is not straightforward to accurately identify associated diagnostic terms and/or conditions in a robust manner using IBM Explorys data.

Here, we develop a general framework that is designed to utilize EHR data (specifically from IBM Explorys) to identify conditions that exhibit *stronger* association with the condition of interest (*intimate partnet violence*) in one population (e.g., older women) as compared to another population (e.g., younger women). We refer to such conditions as *differentially co-morbid*. To address the challenges that stem from the privacy measures of Explorys while pro-viding a robust and easy-to-interpret framework, we: (i) systematically quantify the association of each condition with a target condition of interest (e.g., IPV) in a data-agnostic manner, (ii) compute confidence intervals that take into account the overall rarity of the conditions and the errors to ensure statistical rigor, and (iii) classify the conditions into categories (e.g., high, medium, low prevalence) to provide easy to interpret results. This framework is illustrated in Fig 1.

## Materials & methods

### Data collection

IBM Explorys Therapeutic Dataset provides the *Explorys Cohort Discovery* tool which allows the submission of a query by specifying demographic criteria and/or keywords (for findings or diagnoses) to acquire a subpopulation. As a response, the cohort discovery tool forms a cohort that contains the number of records in the specified subpopulation for each finding and/or diagnosis terms in the database. Throughout this paper, we refer to these diagnoses as *terms*.

We investigate the potential health correlates of IPV in two populations: (i) Older women population of 65+ years of age and Background (BG) population of women 18–65 years of age. We query the *Explorys Cohort Discovery* tool to generate cohorts of interest (provided as S1 Data) corresponding to these two populations (Fig 1a), which are specified as follows:

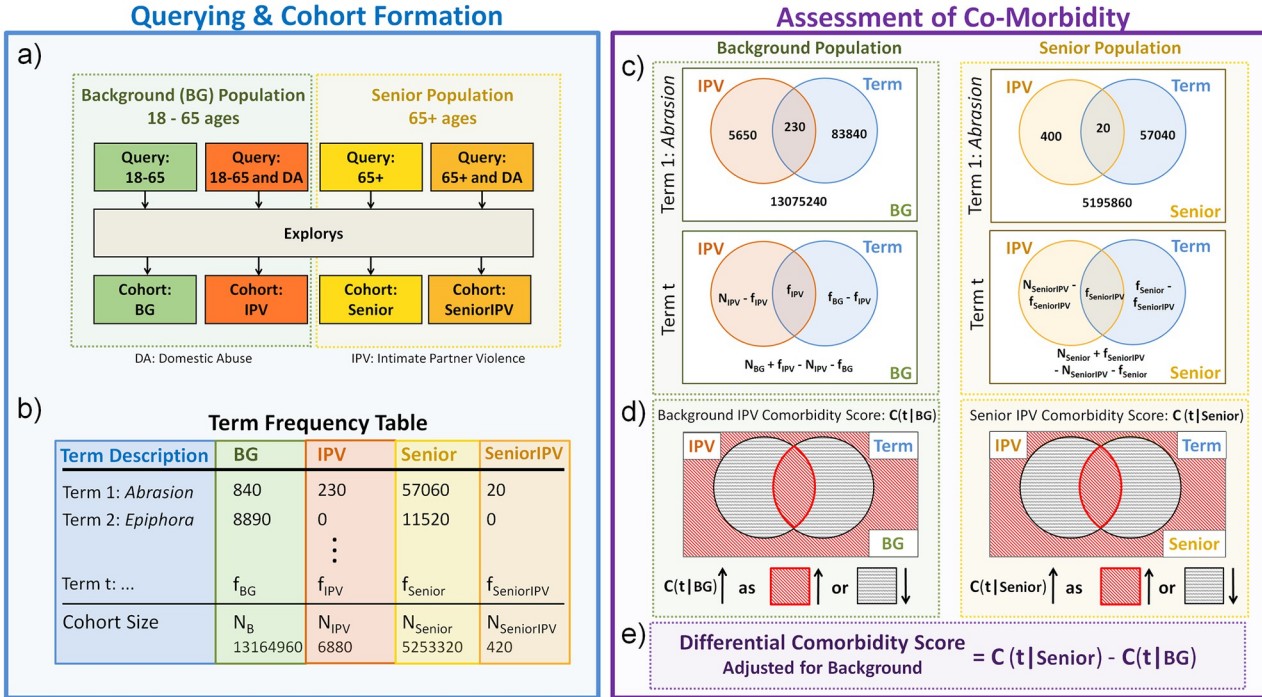

**Fig 1. Flowchart illustrating our pipeline for mining electronic health records to identify health correlates of intimate partner violence against older women.** (a) Generated cohorts for background and older populations. (b) Each cohort contains a frequency table indicating the number of records for each term. (c) 2 × 2 contingency tables (shown as Venn diagrams) are constructed for both background and older women populations and for each term *t*. (d) Using the contingency tables, intimate partner violence (IPV) prevalence scores are computed for both populations. (e) A differential prevalence score is computed for each term to uncover terms that are more associated with IPV in older women population compared to background (BG).

- *BG Cohort*: All records of women 18–65 years of age with a diagnosis of a disease.

- *IPV Cohort*: All records of women 18–65 years of age containing "Domestic Abuse" in the findings field. It correspond to a subpopulation of the BG Cohort having IPV.

- *Senior Cohort*: All records of women 65+ years of age with a diagnosis of a disease.

- *SeniorIPV Cohort*: All records of women 65+ years of age containing "Domestic Abuse" in the findings field. It correspond to a subpopulation of the Senior Cohort having IPV.

## Querying and cohort formation

We ran all queries in June 2019. Each query result (i.e., cohort) $X$ contains the following information: (1) Cohort size $N_X$ indicating the total number of records in $X$, (2) a list of terms $T$ (there are around 18000 terms in the database), and (3) a frequency table $f_X$ that contains for each term $t \in T$ the number of records $f_X(t)$ identified with $t$ (Fig 1b). We provide the frequency tables of all cohorts in S1 Data. To denote the number of records in a population of interest $Z$ (Senior or BG), we use the following notation:

- $N_Z$: Total number of records in population $Z$.

- $N_Z(IPV)$: Number of records in population $Z$ having "Domestic Abuse" as a finding. This number is equal to cohort sizes $N_{IPV}$ and $N_{SeniorIPV}$ respectively for BG and senior populations.

- $N_Z(t)$: Number of records diagnosed with $t$ in population $Z$. This is directly obtained from term frequency table $f_Z$.

## Assessment of co-morbidity

**Constructing contingency tables.** For each population of interest $Z$ (Senior or BG) and term $t$, we construct a $2 \times 2$ contingency table (Fig 1c). This table contains the number of records in $Z$ for all combinations of the existence and absence of IPV and term $t$ variables, i.e.:

- $N_Z(t, \text{IPV})$: Number of records diagnosed with $t$ and has "Domestic Abuse" as a finding. This number is directly obtained from term frequency table of IPV for population $Z$.

- $N_Z(\neg t, \text{IPV}) = N_Z(\text{IPV}) - N_Z(t, \text{IPV})$: Number of records in population $Z$ *not* diagnosed with $t$ but contains "Domestic Abuse" as a finding.

- $N_Z(t, \neg \text{IPV}) = N_Z(t) - N_Z(t, \text{IPV})$: Number of records in population $Z$ diagnosed with $t$ but does *not* contain "Domestic Abuse" as a finding.

- $N_Z(\neg t, \neg \text{IPV}) = N_Z + N_Z(t, \text{IPV}) - N_Z(\text{IPV}) - N_Z(t)$: Number of records in population $Z$ *not* diagnosed with $t$ and does *not* contain "Domestic Abuse" as a finding.

**Computing co-morbidity scores.** For population $Z$ (either senior or background), we consider a term $t$ to be *co-morbid* if, in this population, IPV and term $t$ are significantly more frequently observed *together* rather than separately. We quantify this using the co-morbidity score $C(t|Z)$, which is defined as the log-odds ratio $\text{LOR}(t, \text{IPV}|Z)$:

$$
\begin{aligned}
\text{LOR}(t, \text{IPV}|Z) \quad &= \log_2\left(\frac{N_Z(t, \text{IPV})N_Z(\neg t, \neg \text{IPV})}{N_Z(\neg t, \text{IPV})N_Z(t, \neg \text{IPV})}\right) \\
&= \log_2(N_Z(t, X)) + \log_2((N_Z - N_Z(t) - N_Z(\text{IPV}) + N_Z(t, \text{IPV})) \\
&\quad - \log_2(N_Z(t) - N_Z(t, \text{IPV})) - \log_2(N_Z(\text{IPV}) - N_Z(t, \text{IPV}))
\end{aligned}
\tag{1}
$$

As shown in Fig 1d, $\text{LOR}(t, \text{IPV}|Z)$ increases monotonically as the frequency of term $t$ in $Z \cap \text{IPV}$ subpopulation goes up compared to the frequency of term $t$ in $Z\backslash\text{IPV}$ subpopulation.

**Accounting for variance.** To account for the variability in the estimation of $\text{LOR}(t, \text{IPV}|Z)$, we compute a standard error $SE(t, \text{IPV}|Z)$ as follows:

$$
SE(t, \text{IPV}|Z) = \frac{\sqrt{\frac{1}{N_Z(t, \text{IPV})} + \frac{1}{N_Z(t, \neg \text{IPV})} + \frac{1}{N_Z(\neg t, \text{IPV})} + \frac{1}{N_Z(\neg t, \neg \text{IPV})}}}{\ln(2)}
\tag{2}
$$

Next, we compute $1-\alpha$ level confidence interval as follows:

$$
\begin{aligned}
\text{LOR}_{\min}(t, \text{IPV}|Z) &= \text{LOR}(t, \text{IPV}|Z) - z_\alpha SE(t, \text{IPV}|Z) \\
\text{LOR}_{\max}(t, \text{IPV}|Z) &= \text{LOR}(t, \text{IPV}|Z) + z_\alpha SE(t, \text{IPV}|Z)
\end{aligned}
\tag{3}
$$

where $z_\alpha$ is a critical value obtained from normal inverse cumulative distribution (e.g., $z_\alpha = 1.96$ for $\alpha = 0.05$). Throughout this paper, we use $\alpha = 0.05$ and 95% confidence intervals to determine the statistical significance of the terms.

**Accounting for measurement error due to rounding.** The confidence interval shown in Eq 3 accounts for variance but does not take into account the measurement error due to

rounding of the number of records. For example, if Explorys returns the number of records $N_Z(t, \text{IPV}) = 10$ for a term $t$, this indicates the actual number of records can be anywhere between 5 and 15. For terms with relatively low frequencies, this can potentially alter the log-odds ratio a substantial amount. In order to take the additional uncertainty due to rounding into account, we compute an *augmented confidence interval* using a Monte-Carlo simulation: First, we sample each number of records $N_t$ from $[N_t\text{-}5, N_t\text{+}5]$ uniformly at random and take 100 samples. Next, we use multiple imputation methods [26] to get an estimate of a standard error on LOR that accounts for the rounding of the counts as follows:

1. For each sample $i$ separately, compute the $\text{LOR}_{(i)}(t, \text{IPV}|Z)$ and the corresponding standard error $SE_{(i)}(t, \text{IPV}|Z)$

2. Compute the corrected log odds ratio $\text{LOR}(t, \text{IPV}|Z) = \sum_{i=1}^{100} \text{LOR}_{(i)}(t, \text{IPV}|Z)/100$.

3. Compute within variability $V(t, \text{IPV}|Z) = \sum_{i=1}^{100} SE_{(i)}^2(t, \text{IPV}|Z)/100$.

4. Compute between variability
   $B(t, \text{IPV}|Z) = \sum_{i=1}^{100} \left(\text{LOR}(t, \text{IPV}|Z) - \text{LOR}_{(i)}(t, \text{IPV}|Z)\right)^2/99$.

5. Compute the corrected standard error $SE(t, \text{IPV}|Z) = \sqrt{V(t, \text{IPV}|Z) + \frac{101}{100}B(t, \text{IPV}|Z)}$

The confidence intervals are then computed with standard errors corrected for the rounding of the numbers.

**Accounting for detection bias in health records.** We suspect that EHR databases and hospital records may suffer from a detection bias due to difficulties in diagnosis (e.g., if a severe condition is detected, more medical tests may be performed which can lead to the detection of more terms. Otherwise, there is less attention and many terms go unnoticed). Thus, if not addressed, this bias could lead to an over-estimation in our co-morbidity scores (log-odds ratios) [27]. To help address this issue, we make an estimate $\mu$ of the detection bias on the log odds ratios by looking at the distribution of the co-morbidity scores across all terms (see Fig 2). Based on this estimation, we compute a corrected co-morbidity score $\hat{C}(t|Z)$ that takes

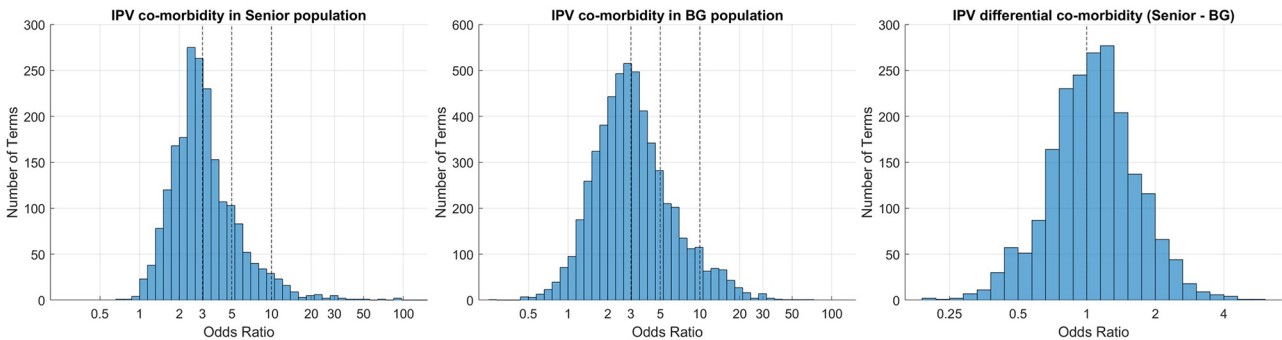

**Fig 2. The distributions of co-morbidity and differential co-morbidity scores in Senior and Background (BG) populations.** (Left & Middle panels) The distribution of the raw co-morbidity scores (i.e., odds ratios) for all terms in Senior and Background populations. The geometric mean of the co-morbidity scores across all terms is used to determine the null level accounting for the detection bias (OR≈3 is the mean across all terms as opposed to the OR = 1 natural level). The null hypothesis thresholds to determine Minor/Moderate/High co-morbidity terms are marked on the histograms (OR = 3/5/10). (Right panel) The distribution of the differential co-morbidity scores for all terms. Since there is not a notable shift in the histogram, OR = 1 is considered as the null level for differential co-morbidity. OR: Odds ratio.

into account a detection bias of level $\mu$ as follows:

$$\hat{C}(t|Z) = C(t|Z) - \mu \tag{4}$$

Thus, we consider OR=$\mu$ as a more appropriate null hypothesis level (as opposed to "OR = 1" natural level) in the presence of a detection bias of magnitude $\mu$. To estimate the magnitude of the detection bias $\mu$, we consider the geometric average of the raw co-morbidity scores over all terms as a guideline (mean odds ratio is respectively: 3.16 and 3.18 for Senior and Background populations, Fig 2). Thus, we consider $\mu = 3$ as the null level indicating no co-morbidity. Here, our reasoning is that if all terms exhibit a strong association (as indicated by the mean co-morbidity score), this association is likely not due to an inherent co-morbidity in the population, but rather is related to record keeping or the detection of the terms (for example, if a patient has a severe condition like IPV, more inquiry and more medical tests may be applied, thus leading to a greater fraction of the terms to be detected).

**Accounting for multiple comparisons.** Our aim in this study is to identify terms with highest co-morbidity and estimate a lower bound on their effect size. For this purpose, we utilize confidence intervals (CIs) and rank the terms according to the lower bound of their interval. However, this process causes a multiple comparisons problem: After such a sorting is applied and top terms are taken, the confidence intervals are no longer valid (not valid in the sense that lower bound of the interval no longer imply statistical significance). To overcome this issue, we aim to bound the false discovery rate (FDR) of the findings (note that we aim to bound FDR as opposed to the family-wise error rate to avoid being overly stringent while determining the significant terms, thus we bound the average number of false discoveries and not the probability of making a false discovery). For this purpose, we utilize the Benjamini-Hochberg (BH) procedure [28] in combination with a more recent work [29] that correct the false coverage rate (FCR) of the confidence intervals for a given selection of significant items obtained from BH procedure (shortly BH-selected FCR corrected CIs). This process goes:

1. Determine a null hypothesis (e.g., OR = 1) and compute p-values.

2. Apply BH-procedure and find signficant terms at $\alpha$ level. Suppose R out of M terms are deemed significant.

3. Correct the CIs of these significant terms, simply by constructing a CI at $1-\alpha R/M$ level (instead of at $1-\alpha$ level).

Here, to avoid specifying a fixed null hypothesis (from which we could only learn that null hypothesis is satisfied or not), we extend this process *for a series of null hypotheses*: OR = $\mu^*$. As the null level $\mu^*$, we practically consider all levels (sampled logarithmically in 0.01 intervals) and apply BH-procedure for each level $\mu^*$. Next, for each term $t$, we find the highest $\mu(t)$ a term would be deemed significant (after BH correction) and construct the corresponding FCR corrected confidence intervals. Note that, the lower bound of these corrected intervals (equal to $\mu(t)$) answers the question:

- "What is the maximum level $\mu^*$ a term $t$ would be deemed significant after correcting for FDR?"

Thus, this allows us to avoid relying on arbitrary significance thresholds (e.g., OR = 10), and allows us to answer questions like "Which terms would no longer be deemed significant if we selected the significance threshold to be OR = 10.1 instead?" simply by looking at the confidence intervals. To summarize, the process that we apply to take into account of multiple comparisons is as follows:

1. Repeat BH-procedure for testing OR = $\mu^*$ for all logarithmically spaced $\mu^*$ with 0.01 intervals (in log2 base).

2. For each term $t$, find the maximum $\mu(t)$ a term t would be deemed significant at $\alpha = 0.05$ level (after the BH-procedure), and take note of the number of terms $R(t)$ that are identified as significant at $\mu(t)$ level.

3. Sort the terms in descending order according to $\mu(t)$.

4. For each term, construct confidence intervals at $1 - \alpha R(t)/M$ level that are corrected to bound FDR (for the most significant top $k$ terms for any $k$). Since the distribution of log-odds ratio is symmetric and we know the lower bound $\mu(t)$, the upper bound of this interval can also be obtained from $2^{(2\text{LOR}(t,\text{IPV}|Z)-\log_2\mu(t))}$.

## Assessment of differential co-morbidity

One of the objectives of this study is to uncover terms that are frequently observed together with IPV in older women population more so than the background population. To this end, we compute a *differential co-morbidity score* DC($t$) (Fig 1e) compared to the background (BG) population:

$$\text{DC}(t) \quad = \text{C}(t|\text{Senior}) - \text{C}(t|\text{BG}) = \text{LOR}(t, \text{IPV}|\text{Senior}) - \text{LOR}(t, \text{IPV}|\text{BG}) \tag{5}$$

Since the older women and background populations are independent, the standard error of the differential co-morbidity for term $t$ can be computed as follows:

$$\sigma(t) \quad = \sqrt{SE^2(t, \text{IPV}|\text{Senior}) + SE^2(t, \text{IPV}|\text{BG})} \tag{6}$$

Using the standard error, we compute the corrected confidence intervals for the differential co-morbidity as detailed in *"Accounting for multiple comparisons"* section. Overall, we consider a term $t$ to be differentially co-morbid with *high* confidence if $\text{DC}_{\min}(t)$ is greater than zero. Otherwise, we conclude that it has a low confidence level to make a judgement.

## Experimental setting

The datasets obtained from IBM Explorys system contain information about a total of 18863 terms. We assess the co-morbidity of these terms with IPV in the older women population as well as the background (BG) population. For each term $t$, we compute co-morbidity scores C($t$|Senior), C($t$|BG) and the differential co-morbidity score DC($t$) for the difference between senior and background populations. For each co-morbidity score, we compute the corresponding 95% augmented confidence intervals (corrected to bound the false discovery rate) to assess the statistical significance. We consider a co-morbidity score to be *invalid* if the confidence interval does not have a finite range (e.g., when the term frequency is zero in one or more cohorts).

To make the interpretation of the results easier, we consider three null levels for the co-morbidity scores (OR = 3/5/10) and label the significant findings (at $\alpha = 0.05$ level after accounting for FDR via BH-procedure) as respectively Minor/Moderate/Highly co-morbid terms. Note that, for Minor co-morbidity, the null level is selected to be OR = 3 (as opposed to OR = 1) to take into account the detection bias.

## Results

### Medical terms that are co-morbid in older victims of IPV

We identify 2057 and 5464 valid terms for older women (senior) and background populations respectively (2039 of these are valid for both). The difference in the number of valid terms is likely due to the difference in cohort sizes as the background population has around 2.5 times more number of records than the older women population (13164960 vs. 5253320).

First, we start by investigating the terms that are statistical significant (for null hypothesis OR = 1, $\alpha$ = 0.05, after FDR is bounded using BH-procedure) and we observe a rather interesting result: In both populations, almost all valid terms are deemed statistically significant (4664 terms for background, and 1681 terms for senior population). To investigate whether this is a result of a detection bias in our dataset, we examine the distribution of the co-morbidity scores across all terms in Fig 2. As it can be seen, the distribution of the co-morbidity scores are considerably shifted to the right and are approximately centered around OR = 3 for both senior and background populations (geometric mean for the odds ratio is respectively: 3.16 [1.42, 5.66] and 3.18 [1.84, 4.79] for senior and background populations). This suggests that OR = 3 can be considered as a more natural null level for assessing the co-morbidity with IPV in this dataset, explaining why there are so many significant terms when tested for OR = 1. Note that, we do not observe any notable shift in the distribution of the differential co-morbidity scores (Fig 2 right panel) since the effect of the bias seems approximately equal for senior and background populations which cancels out when we take the difference.

Overall, when we look at the terms with high co-morbidity, we identify respectively 199, 64 and 13 terms with minor, moderate and high co-morbidity in the senior population (and 905, 420 and 165 terms in the background population). Here, we mainly focus on the highly co-morbid terms in the senior population and report the top 20 terms in Table 1 sorted by the minimum bound of their 95% confidence intervals (after they are corrected to bound the false coverage rate). We provide the remaining terms identified as co-morbid in S2 Data. For each term, we provide both the raw co-morbidity scores and their corrected versions where the expected portion of the association due to detection bias is removed (by dividing the raw co-morbidities to OR = 3).

In Fig 3, we visualize the significant findings and compare their co-morbidities in senior and background population. We observe that while most of the terms that are highly co-morbid in senior population are also highly co-morbid in background population, there are some terms with notably higher co-morbidity in the senior population. Next, we focus on such terms exhibiting differential co-morbidity.

### Terms with a higher co-morbidity in older victims of IPV compared to younger victims

We find that there are 162 terms with significant differential co-morbidity (exhibiting higher association with IPV in the older women population). Since there is a large number of findings and these consist of many similar terms (e.g., there is a term for "severe depression" and another for "severe major depression"), we manually annotate and group these based on their general categories and report a few selected term from each category in Table 2. Note that, while making this selection, we take into account of borderline cases by looking at their confidence intervals and also consider the overall co-morbidity of the terms in background and senior populations. Here, we mainly focus on the terms that exhibit significant co-morbidity in senior population in addition to being deferentially co-morbid (corresponding to upper right side in Fig 3). We provide the remaining terms and their assigned categories in S3 Data.

**Table 1. Top 20 terms having high co-morbidity with IPV in older women population.** For each term, we report both the raw co-morbidity scores (the odds ratios) and their corrected versions to alleviate the detection bias. For each co-morbidity score, we report the following: The point estimate (odds ratio), 95% augmented confidence interval corrected to bound FDR, and the corresponding co-morbidity level. The co-morbidity levels are abbreviated: H for high, M for moderate, and *m* for minor. The terms are sorted in descending order according to the minimum bound of their confidence intervals $OR_{min}(t, IPV|Senior)$. This table reports the top 20 terms that exhibit high ($OR \gg 10$) or moderate level of association ($OR \gg 5$) with IPV in the older women population. See S2 Data for a full list of terms identified as significant. OR: Odds ratio, IPV: Intimate partner violence.

|   | Term Description | Co-morbidity in Senior population | | Number of Records | | | |
|---|---|---|---|---|---|---|---|
|   |   | Raw | Corrected for detection bias | BG | IPV | Senior | SeniorIPV |
| 1 | History of abuse | 91.4 [27.9, 299.4] H | 30.5 [9.3, 99.8] H | 36660 | 310 | 2780 | 20 |
| 2 | Maltreatment syndromes | 194.7 [27.9, 1358.4] H | 64.9 [9.3, 452.8] H | 4390 | 100 | 660 | 10 |
| 3 | Poisoning caused by anticonvulsant | 50.4 [16.2, 156.6] H | 16.8 [5.4, 52.2] H | 26920 | 190 | 5100 | 20 |
| 4 | Poisoning caused by sedative AND/OR hypnotic | 46.3 [15.5, 138.3] H | 15.4 [5.2, 46.1] H | 28660 | 200 | 5650 | 20 |
| 5 | Continuous acute alcoholic intoxication in alcoholism | 85.8 [13.7, 535.5] H | 28.6 [4.6, 178.5] H | 7870 | 120 | 1400 | 10 |
| 6 | Continuous opioid dependence | 38.9 [12.9, 116.9] H | 13.0 [4.3, 39.0] H | 25590 | 140 | 6750 | 20 |
| 7 | Chronic post-traumatic stress disorder | 28.2 [12.1, 65.8] H | 9.4 [4.0, 21.9] H | 136510 | 680 | 14260 | 30 |
| 8 | History of physical abuse | 64.6 [10.8, 386.9] H | 21.5 [3.6, 129.0] H | 29000 | 290 | 1990 | 10 |
| 9 | Poisoning caused by central nervous system drug | 21.7 [10.8, 43.6] H | 7.2 [3.6, 14.5] H | 150480 | 820 | 25370 | 40 |
| 10 | Acute drug intoxication | 29.9 [10.5, 85.0] H | 10.0 [3.5, 28.3] H | 69550 | 420 | 8690 | 20 |
| 11 | Alcohol intoxication | 29.5 [10.5, 82.6] H | 9.8 [3.5, 27.5] H | 68940 | 420 | 8650 | 20 |
| 12 | Contusion of multiple sites | 20.8 [10.5, 41.1] H | 6.9 [3.5, 13.7] H | 55570 | 850 | 26750 | 40 |
| 13 | Pathological drug intoxication | 29.4 [10.3, 84.0] H | 9.8 [3.4, 28.0] H | 70250 | 430 | 8860 | 20 |
| 14 | Posttraumatic stress disorder | 22.1 [10.0, 48.7] M | 7.4 [3.3, 16.2] M | 161610 | 720 | 17980 | 30 |
| 15 | Toxic effect of ethyl alcohol | 27.3 [9.8, 75.7] M | 9.1 [3.3, 25.2] M | 73690 | 450 | 9480 | 20 |
| 16 | Poisoning caused by chemical substance | 18.5 [9.5, 36.3] M | 6.2 [3.2, 12.1] M | 154130 | 730 | 29160 | 40 |
| 17 | Poisoning caused by psychotropic agent | 26.4 [9.5, 73.8] M | 8.8 [3.2, 24.6] M | 60390 | 360 | 9730 | 20 |
| 18 | Alcohol abuse | 16.7 [9.3, 29.9] M | 5.6 [3.1, 10.0] M | 220590 | 1200 | 42040 | 50 |
| 19 | Drug abuse | 19.8 [9.1, 43.0] M | 6.6 [3.0, 14.3] M | 190400 | 1020 | 20100 | 30 |
| 20 | Nondependent alcohol abuse | 23.6 [8.7, 63.7] M | 7.9 [2.9, 21.2] M | 36180 | 230 | 11090 | 20 |

## Discussion

Much of the past research on IPV is based on data from younger women. However, recent studies demonstrated that the older women in growing numbers are also often victims of physical and nonphysical forms of IPV (e.g. emotional, psychological and economic abuse) [7, 13, 14, 30, 31]. Our aim in this study was to investigate the health correlates of IPV among older women. We presented a general framework that is designed to utilize electronic health record (EHR) data to identify health correlates of a condition of interest (e.g., IPV) that is specific to a target population (e.g., older women). We mined the EHR data that is available through IBM Explorys, a database containing demographic and diagnostic information gathered from diverse institutions across the United States. The data is analyzed by systematically assessing associations of medical terms, computing confidence intervals that take into account the rounding errors, and classifying the terms into confidence levels.

Based on the analysis in our previous study [27] and looking at the distributions of the co-morbidity scores (in Fig 2), we reason that the over-population of the terms deemed as significant using the standard approach (i.e., testing for *OR* = 1 at 0.05 level accounting for FDR) is likely because of a detection bias in the health records. The presence of a severe condition like IPV would naturally warrant more exploration during the screening and this can result in more terms to be detected (including those that would otherwise undetected). Thus, this can cause an artificial association with IPV that is not representative of the inherent population.

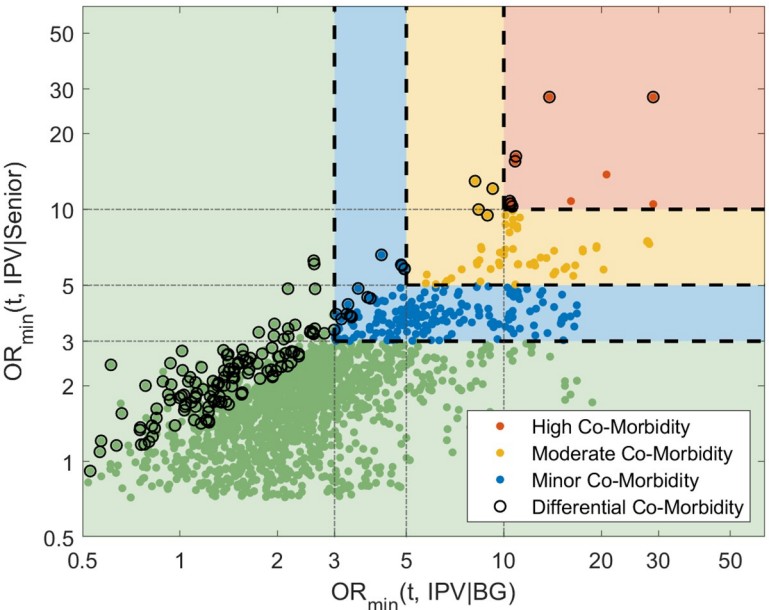

**Fig 3. X-axis and Y-axis indicate the minimum bounds of 95% augmented confidence intervals (corrected to bound the FDR) for IPV co-morbidity scores in Senior and Background (BG) populations i.e., OR(t, IPV|Senior), and OR(t, IPV|BG) respectively.** The terms identified with High/Moderate/Minor co-morbidity in both populations are shown in red/yellow/blue regions respectively. The terms identified as differentially co-morbid (having significantly higher co-morbidity in Senior population) are marked with black circles. OR: Odds ratio, IPV: Intimate partner violence.

Our initial analyses regarding the co-morbid terms in Senior population indicated that substance abuse and poisoning associated with substances are significantly co-morbid with IPV in older women (Table 1). This finding is particularly strong in that 17 of the top 20 terms that are co-morbid with IPV are substance abuse related, while the remaining 3 are directly associated with abuse (history of abuse, maltreatment syndromes, and history of physical abuse). It is important to note that screening for substance abuse and medication overuse among older women with a history of IPV is critical since these terms are highly correlated.

In contrast to terms with significance co-morbidity with IPV, terms with significant differential co-morbidity with IPV (in older women as compared to the background population) were more diverse (Table 2). Specifically, we identified 161 diagnostic terms that exhibited a significantly stronger association with IPV in older women as compared to the background population. These terms included history of abuse, those related to mental health (21 terms) and substance use issues (5), neoplasm, tumors and growths (26 terms), musculoskeletal issues (25 terms), disorders (20 terms), skin problems (11 terms), ear, nose and throat issues (11 terms), inflammation (7), neurological conditions (6), immune problems (5), women's health (OB-GYN) (5 terms), infectious disease (4 terms), procedures (4 terms), eye disease (3 terms), drug interactions (3 terms), acute conditions (2 terms) and other conditions (3 terms).

Our detailed analysis indicated that mental health conditions such as major depression in partial remission, adjustment disorder with mixed emotional features, chronic post-traumatic stress disorder, anxiety disorder, mood disorder are more likely to occur among older women who have been abused by their partners as compared to younger women. Also continuous opioid dependence, and alcohol intoxication were also found to be differentially co-morbid with IPV in older women as compared to the background population. Past research reports that IPV is associated with an increased likelihood of clinical depression and suicide attempts

**Table 2. Terms that exhibit higher co-morbidity with IPV in older women population compared to the background (BG) population.** For each term, we provide the co-morbidity scores for Senior and BG populations (after correcting for detecting bias), and the differential co-morbidity score. For each co-morbidity score, we report the following: The point estimate (odds ratio), 95% augmented confidence interval corrected to bound FDR, and the corresponding co-morbidity level. The co-morbidity levels are abbreviated: H for high, M for moderate, and *m* for minor. All 20 reported terms are identified as differentially co-morbid with high confidence (this indicates that they exhibit higher association with IPV in older women population compared to the BG population in a statistically significant manner). See S3 Data for a full list of terms identified as significant and their assigned categories. OR: Odds ratio, IPV: Intimate partner violence.

| Category | Term Description | Corrected Co-morbidity in Population Z | | Differential Co-morbidity | Number of Records | | | |
|---|---|---|---|---|---|---|---|---|
| | | Senior | Background (BG) | Senior vs. BG | BG | IPV | Senior | SeniorIPV |
| Drug Interactions | Poisoning caused by anticonvulsant | 16.8 [5.4, 52.2] H | 4.7 [3.6, 6.0] H | 3.61 [1.27, 10.25] | 26920 | 190 | 5100 | 20 |
| Miscellaneous | History of abuse | 30.5 [9.3, 99.8] H | 5.7 [4.6, 7.0] H | 5.38 [1.58, 18.32] | 36660 | 310 | 2780 | 20 |
| Substance Use Issues | Continuous opioid dependence | 13.0 [4.3, 39.0] H | 3.6 [2.7, 4.7] M | 3.64 [1.26, 10.51] | 25590 | 140 | 6750 | 20 |
| | Alcohol intoxication | 9.8 [3.5, 27.5] H | 4.1 [3.5, 4.9] H | 2.38 [1.02, 5.55] | 68940 | 420 | 8650 | 20 |
| Mental Health | Chronic post-traumatic stress disorder | 9.4 [4.0, 21.9] H | 3.5 [3.1, 4.0] M | 2.68 [1.24, 5.82] | 136510 | 680 | 14260 | 30 |
| | Major depression in partial remission | 5.2 [2.1, 12.8] M | 1.2 [0.9, 1.7] | 4.30 [1.31, 14.15] | 31750 | 60 | 16870 | 20 |
| | Adjustment disorder with mixed emotional features | 3.9 [1.6, 9.5] *m* | 1.1 [0.9, 1.4] | 3.62 [1.25, 10.44] | 77440 | 130 | 22080 | 20 |
| | Generalized anxiety disorder | 1.9 [1.3, 2.8] *m* | 1.1 [1.0, 1.2] *m* | 1.70 [1.10, 2.63] | 578410 | 910 | 211620 | 80 |
| | Chronic mood disorder | 1.9 [1.3, 2.9] *m* | 1.2 [1.1, 1.4] *m* | 1.55 [1.01, 2.39] | 473480 | 830 | 176180 | 70 |
| | Anxiety disorder | 1.7 [1.2, 2.3] *m* | 1.2 [1.1, 1.3] *m* | 1.40 [1.02, 1.92] | 1718770 | 2420 | 624790 | 170 |
| Muscle Skeletal Issues | Injury of ligament of hand | 4.9 [1.9, 12.2] M | 2.0 [1.6, 2.3] *m* | 2.48 [1.02, 5.98] | 90940 | 270 | 17730 | 20 |
| | Synovitis | 1.8 [1.0, 3.0] *m* | 0.8 [0.6, 0.9] | 2.32 [1.19, 4.55] | 188300 | 220 | 101300 | 40 |
| | Acquired deformity of joint of foot | 1.4 [0.9, 2.2] | 0.5 [0.5, 0.7] | 2.64 [1.31, 5.33] | 176850 | 150 | 159380 | 50 |
| Disorders | Hypoglycemia | 2.1 [1.1, 3.8] *m* | 1.1 [0.9, 1.3] | 1.96 [1.00, 3.86] | 104830 | 170 | 64730 | 30 |
| | Developmental disorder | 2.3 [1.0, 5.0] *m* | 0.8 [0.7, 0.9] | 2.86 [1.12, 7.30] | 425210 | 510 | 36860 | 20 |
| | Nutritional disorder | 1.0 [0.8, 1.3] | 0.5 [0.5, 0.5] | 2.01 [1.31, 3.10] | 1289750 | 950 | 980750 | 170 |
| | Vitamin D deficiency | 0.9 [0.7, 1.2] | 0.5 [0.4, 0.5] | 1.99 [1.28, 3.10] | 911940 | 640 | 669660 | 120 |
| Skin Problem | Tinea pedis | 2.4 [1.1, 5.5] *m* | 0.8 [0.6, 1.0] | 3.22 [1.15, 9.03] | 67660 | 80 | 34850 | 20 |
| Infectious disease | Infectious disease of lung | 1.9 [1.1, 3.2] *m* | 0.9 [0.7, 1.1] | 2.09 [1.08, 4.03] | 78290 | 110 | 96360 | 40 |
| Women's Health | Pelvic injury | 1.8 [1.1, 3.0] *m* | 1.0 [0.9, 1.1] | 1.76 [1.01, 3.07] | 612530 | 890 | 102340 | 40 |
| Neurological | Migraine | 1.6 [1.1, 2.4] *m* | 0.9 [0.8, 1.0] | 1.80 [1.12, 2.87] | 1122900 | 1390 | 208030 | 70 |
| Neoplasm/Tumor | Neoplasm of stomach | 1.7 [0.8, 3.6] | 0.4 [0.2, 0.6] | 4.75 [1.21, 18.72] | 35500 | 20 | 51160 | 20 |
| ENT issues | Posterior rhinorrhea | 1.4 [0.7, 2.9] | 0.5 [0.4, 0.6] | 3.10 [1.12, 8.56] | 96150 | 70 | 59130 | 20 |
| Inflammation | Pharyngitis | 1.3 [0.9, 1.8] | 0.7 [0.7, 0.8] | 1.75 [1.10, 2.78] | 1893860 | 1830 | 263480 | 70 |

among women in general [11]. A systematic exploration of the predominant mental health conditions of older women abuse and psychological well-being demonstrates that depression, anxiety, and post-traumatic stress disorder are among prevalent problems [32]. Also, in-depth interviews conducted with abused women aged from 63 to 79 found that older abused women are more prone to symptoms related to mental health issues like anxiety, depression and negative view of self [33]. Similarly, clinical and case-controlled studies indicated poor mental health, particularly depression and dementia as common problems in geriatric clinics among abused older people [34]. It is possible that these conditions are partially related to the isolation of older adults [35, 36]. Moreover, the isolation coupled with IPV likely makes older people more prone to depression. Specifically, we observe that older victims of IPV suffer from major depression roughly 4 times more than younger women (95% confidence interval: [1.35, 11.71]).

Findings also indicated that musculoskeletal issues such as acquired deformity of joint of foot, acquired deformity of the lower limb, injury of ligament of hand, flexion deformity,

polyarthropathy are terms that are more prevalent in the older IPV victims population as compared to the background population (Table 2). This is also consistent with prior research reporting that older adults may come into the emergency department due to fall injuries that could be linked to IPV. Although older women are naturally prone to musculoskeletal issues and osteoporosis resulting in loss of mobility and physical independence, this rate is even higher among older women with a history of IPV. It is possible that a physical trauma as a result of IPV may negatively impact the already vulnerable musculoskeletal system through scaring from an injury, inflammatory disease or hyperglycemia, which we also observed more frequently among older women with a history of abuse, doubling the risk of muscle musculoskeletal issues that are functionally limiting and physically debilitating [37–39]. Injuries are more common among terms that are more co-morbid in the older women population as compared to the background population (Table 2). This is also consistent with prior research reporting that older adults may come into the emergency department due to fall injuries that could be linked to IPV [5]. A Nationwide Emergency Department Sample from 2006 to 2009 revealed that there were approximately 28,000 ED visits per year due to IPV [40]. Older adults, in particular, may come into the ED due to fall injuries that could be linked to IPV [6, 12]. Therefore, the emergency department (ED) provides a valuable opportunity to identify and treat this at-risk population [6].

Another important category that emerged as differentially co-morbid with IPV in older women was neoplasms and tumors, with neoplasm of stomach showing significant differential co-morbidity. Past researchers including Cesario et al. [41] interviewed three hundred abused women to explore the link between cancer and IPV. They found that abused women reported 10 times higher levels of a diagnosis of cervical cancer than the general population. Past research also suggested a link between breast cancer and IPV [42]. Researchers also discovered that cancer patients with history of IPV were twice more likely to develop estrogen and/or progesterone negative tumor receptors than patients without IPV history [42]. As concordant with our mental health related findings (Table 2), IPV is frequently linked with higher levels of perceived stress, PTSD and depression [43]. These conditions have been thought to be linked to cancer progression by mediating the link through increasing the vulnerability through smoking, alcohol consumption, and obesity. Furthermore, cancer survivorship is negatively affected by IPV through delays in screenings, diagnosis and treatment as well as women's ability to cope with and recover [44]. Consequently, while our finding on the high differential co-morbidity of neoplasm of stomach is a new finding that is not reported in the literature, there is strong support for multiple links between IPV and other cancers that warrants further investigation of the relationship between IPV and neoplasm of stomach in older women.

The generality of the diagnostic terms affecting multiple organ systems demonstrates the importance of Family Health and Wellness Clinics and Women's Health Clinics as critical fields to detect IPV. In addition, the basic routine health care visits for most women are critical for a first line of defense against more serious IPV-related injury and ailment, especially considering the finding that 84% of women who confide in someone about the abuse choose to tell their health care provider [45]. Past research also indicated that as older people need longer recovery time, health outcomes of abuse in later life could be more overwhelming [46]. Furthermore, as one of the front lines of treatment, the ED provides a safe environment for older adult victims to seek help. It can also serve as a point of contact for the effective distribution of referral information, as health care professionals have unique access in the ED to otherwise hard-to-reach victims [6, 9]. However, ED screening has some limitations. For example, screening windows for IPV in the ED may be too brief to determine the extent, forms and the effects of the IPV. It can also be difficult to conduct interventions in such a time-sensitive and public environment [17, 47]. Findings of this study reinforce the necessity to have screening

measures at place in the emergency department for women of all ages, regardless of whether they present with trauma injuries. The high percentage of women who suffer emotional and physical abuse makes it imperative that interventions exist for women with history of IPV.

## Limitations

As discussed in the Introduction, there are multiple barriers to the reporting and identification of IPV in all women, which may be accentuated for older women. While one motivation for this study is to identify potential markers of IPV to help overcome these barriers, it is important to note that these barriers also impose limitations on the data we analyze in this study. To be more specific, the cases of IPV that are reported in the EHR database can be subject to detection bias, e.g., more severe cases of IPV may be over-represented in our cohorts. Thus the associations we identify here may be associated with severe IPV as opposed to more common forms of IPV and emotional abuse.

Another limitation of our findings is that they do not provide causal interpretations of the associations that are identified. Since the data is cross-sectional and is provided in summaries (i.e., no sample-specific data is available), our findings are limited to high-level associations. Since sample-specific data is not available, we are not able to perform cluster analyses or develop supervised models that can be test with cross-validation, posing limitations to the interpretation and validation of our findings. For these reasons, dedicated data collection efforts that target specific populations, take into account longitudinal patterns, and investigate causal relationships are needed to further characterize the mechanisms of these associations. Our findings can provide useful starting points for such studies.

## Conclusion

In conclusion, this study investigates the medical conditions (terms) that are associated with IPV in older women. There are many potential factors that may contribute to the increasing rates of reported violence amongst the older adult population. Clinicians must be aware of IPV for proper care of older adult patients, especially for those with suspicious symptoms. We expect that terms that are identified in this study could be useful for screening IPV in older women and facilitate timely interventions. Furthermore, the prevalence estimations provided in this study could give insight about the risk of IPV in both older and younger women populations. Evaluations on IPV could be conducted on all women that present to the Health Care System including the emergency department settings, family medicine department settings, women's health clinics, and nursing homes or retirement communities. Such efforts can lead to reduced recurrence of violence, improved mental health and overall higher quality of life among this vulnerable population.

## Supporting information

**S1 Data.**
(XLSX)

**S2 Data.**
(XLSX)

**S3 Data.**
(XLSX)

**S1 File.**
(ZIP)

## Author Contributions

**Conceptualization:** Serhan Yılmaz, Erkan Gunay, Da Hee Lee, Kathleen Whiting, Mehmet Koyuturk, Gunnur Karakurt.

**Data curation:** Serhan Yılmaz, Gunnur Karakurt.

**Formal analysis:** Serhan Yılmaz, Mehmet Koyuturk.

**Funding acquisition:** Mehmet Koyuturk, Gunnur Karakurt.

**Investigation:** Serhan Yılmaz, Erkan Gunay, Mehmet Koyuturk, Gunnur Karakurt.

**Methodology:** Serhan Yılmaz, Mehmet Koyuturk, Gunnur Karakurt.

**Project administration:** Kathleen Whiting, Gunnur Karakurt.

**Resources:** Kristin Silver, Gunnur Karakurt.

**Supervision:** Erkan Gunay, Mehmet Koyuturk, Gunnur Karakurt.

**Validation:** Serhan Yılmaz, Erkan Gunay, Da Hee Lee, Gunnur Karakurt.

**Visualization:** Serhan Yılmaz, Mehmet Koyuturk.

**Writing – original draft:** Serhan Yılmaz, Erkan Gunay, Da Hee Lee, Kathleen Whiting, Kristin Silver, Mehmet Koyuturk, Gunnur Karakurt.

**Writing – review & editing:** Serhan Yılmaz, Kathleen Whiting, Kristin Silver, Mehmet Koyuturk.

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
