## [Decision Letter · Decision Letter 0]

9 Nov 2021

PONE-D-21-21436Identifying Adverse Health Correlates of Intimate Partner Violence against Older Women: Mining Electronic Health RecordsPLOS ONE

Dear Dr. Karakurt,

Thank you for submitting your manuscript to PLOS ONE. After careful consideration, we feel that it has merit but does not fully meet PLOS ONE’s publication criteria as it currently stands. Therefore, we invite you to submit a revised version of the manuscript that addresses the points raised during the review process.

We look forward to receiving your revised manuscript.

Kind regards,

Astrid M. Kamperman

Academic Editor

PLOS ONE

Journal Requirements:

"Gunnur Karakurt (PI) This publication was made possible by R01-LM012518 from the National Library of Medicine. Its contents are solely the responsibility of the authors and do not necessarily represent the official views of the NIH" 

Reviewers' comments:

Reviewer's Responses to Questions

**Comments to the Author**

1. Is the manuscript technically sound, and do the data support the conclusions?

Reviewer #1: Partly

Reviewer #2: No

2. Has the statistical analysis been performed appropriately and rigorously? 

Reviewer #1: I Don't Know

Reviewer #2: No

3. Have the authors made all data underlying the findings in their manuscript fully available?

Reviewer #1: Yes

Reviewer #2: Yes

4. Is the manuscript presented in an intelligible fashion and written in standard English?

Reviewer #1: Yes

Reviewer #2: Yes

5. Review Comments to the Author

Reviewer #1: Thanks to the authors for taking the time and effort to prepare this article. Here are some comments that the authors can consider for revising their paper.

-The mathematical notations used in the article seemed unnecessary in my opinion, an example: the explanation how contingency tables were formed. I suggest authors to find simple analytic language to express their approach concisely and avoid notations when not needed.

I am confused why a log odds ratio was used to present prevalence.

-This does not seem to be a statistical methods paper; I wonder why readily available statistical software were not used to do the analysis, but rather manual calculation was preferred. For example, do the authors think using a software-based analysis would have saved them the space and effort for the section “Accounting for measurement error due to rounding” and made it easier for the readers to focus on the findings more?

-The result section is not elaborate enough. The section mostly refers to tables and does not point out to key and important findings from the work at all.

-The limitation section in discussion should list limitations of the study and not limitations in the topic (those can go in the introduction)

-The conclusion should pinpoint which symptomatology or diagnoses the authors think could be flags for IPV in older women, which is missing. Rather general statements were made.

Reviewer #2: PLOS One

referee report

PONE-D-21-21436: "Identifying Adverse Health Correlates of Intimate Partner Violence against Older Women: Mining Electronic Health Records"

2021 11 07

First, note that this reviewer is a biostatistician and not a subject matter reviewer. Therefore, most of my review is concerned with the statistical methods used in the analysis.

Overview: This paper looks at the relationship between a large number of possible factors and intimate partner violence (IPV) among seniors. The data come from a large data source and the investigators identify a large number of potential factors that are looked at as possibly being related to IPV. (Initially, there are >18K factors considered.) For each examined factor, the authors basically construct a 2x2 table that is standard in epidemiological research. That is, this table looks at the cases versus the controls which are those who are and are not identified as victims of IPV. For each group (case and control), one counts the number who are exposed and not exposed to the individual factor. After creating a 2x2 table for each factor, they compute the log of the odds ratio and the associated confidence interval. If the confidence interval is greater than 0 (so the odds ratios are bigger than 1), the investigators then basically declare the factor important. Also, the reviewers compare the odds ratio for the senior population to the odds ratio of the non-senior population to see if the effect of each of the factors are discernible higher for the senior population. In the main body of the paper, the authors list some of these top factors. There is mention of a supplementary table with a larger list of the factors which are identifies as important. Note: This reviewer does note that there were some other subtle issues that are addressed in the paper. The above overview is this reviewer’s understanding of the main concepts in the paper.

Overall, I like the general idea of the paper. I think that electronic health records are a largely untapped source of valuable information. However, I have important concerns that this manuscript does not do enough to account for the high number of multiple tests performed for this analysis. In the statistical literature, this is known as the multiple testing problem. Although the authors don’t mention that they are doing formal hypothesis testing, the essence of this problem still applies to this study. It might seem that I am suggesting substantial changes in the analysis; however, they would probably not take long to do.

I would further note, that I did look up some papers that were produced from this general research group. Many of my below comments would have been answered in a satisfactory manner if they used the methods that they used in some other papers. I would specifically point out the paper: Karakurt, Patel, Whiting, Koyuturk, 2017, J Fam Viol 32:79-87, DOI 10.1007/s10896-016-9872-5. I will be referring to this paper as Karakurt et al in the below.

I will first list some of the statistical notes/comments/issues and then list more general matters.

1) Multiple testing. Using the criteria that a factor is acceptable in some sense (high or medium confidence) if the confidence interval is greater than 0 is basically mathematically equivalent to a null hypothesis test. (Technically, this would be a 1-way test at the .025 level.) So, the authors are doing 18k null hypothesis test (or, considering only the "valid" test, there are 2056 different test). This is the classical multiple testing problem. You are saying that something is important if the test statistic is would be rare if it was random. That is, rare means that the outcome only occurs with about a 5% probability. Then you run this 2000 or so times and you see lots of “rare” things. This is a classic problem and there are many different ways to approach this.

a) In the other paper (Karakurt et al), a Bonferroni correction is made. So, the authors are aware of the problem. When one is doing many test, this method might be too conservative.

b) As an alternative to the Bonferroni, I prefer using the False Discovery Rate procedure (reference: Benjamini and Hochberg 1995 and Benjamini and Yekutieli 1999.) This reviewer is a fan of this method. At the end of the analysis, one gets a set of factors. In this set, it is acknowledge that some might still be by chance but overall the set contains factors which are highly likely to be important.

2) The issue that the observations are only know to the nearest "tenth". Here, the authors use a very conservative correction. When creating the log odds ratio (LOR) and the standard error estimate, they assume that the actual value is the worse possible. That is, the counts are rounded off to the nearest 10.

a) In the other paper (Karakurt et al), the authors use a Monte Carlo simulation method to sample the possible values which would be the observed value plus or minus 5. That seems very reasonable to me. Basically, this approach is a type of multiple imputation used for missing data. (Also note, the authors in that paper then compute an adjusted confidence interval. This adjusted confidence interval can be used to approximate an adjusted standard error for the LOR.)

b) In the other paper (Karakurt et al), take the sampled values are used to create an estimated confidence interval. Instead of that, one option is that one can use multiple imputation methods. That is, take the sampled values and then use multiple imputation methods to get an estimate of the standard error of the LOR which accounts for the interval censoring of the data. (see for example equations 14.7-14.10 in this article: https://www.sciencedirect.com/topics/mathematics/multiple-imputation for one reference for these formulas. These formulas are available many other places.)

c) Instead of random sampling, one can do some calculations based on the fact that one is sampling from finite discrete distribution. That is, if one is planning on sampling an integer between -5 and 5, then just calculate the LOR and standard error for each of these 11 points and then use the formulas in (b) above.

3) Prevalence: The term prevalence has a well-defined definition in the epidemiology and population health sciences. It is the number of events in a population at a designated time. There is a similar definition of the prevalence rate. The use of the term "prevalence score" to mean the log of the odds ratio is very confusing and is a misuse of well-established terms.

4) Difference between the log odds ratios between the senior population and the younger population. On page 4, the authors look at “Assessment of Differential Prevalence”. They define this term in equation 5 and then give their formula for the confidence intervals in equation 6. This is a very reasonable thing to do. It is the difference between the two LOR. It is the increase in the odds between the senior population versus the non-senior population. This is a very common statistics to look at therefore the statistical properties are well known. The authors need to change the way that they compute these confidence intervals.

Note the following statistical properties: a) the LOR statistics are approximately normally distributed with the standard errors as given in the equation 2. b) Since the senior population and the younger population are completely different populations, then these samples are statistically independence. c) Therefore, the difference of the two statistic defined in equation 5 has an approximate a normal distribution. d) standard error of difference of between the sampled log odd ratio's between these two populations is just "square root of the sum of the squared standard errors". That is, if the standard errors are S1 and S2, then the standard error of the differences is sqrt( S1^2 + S2^2). One can then get the confidence interval of the difference in the log odds ratio by the methods already discussed in the paper. (That is, it is the difference +/- 1.96* standard error of the difference as described in equation 3).

5) I don't see what the value of equation 4. That is, the value of using a mean adjusted prevalence score which is obtained by subtracting the LOR by the average of the LOR and then using that to declare "high" effect. That seems like an attempt to control for the multiple testing, but does not appear to have any validity in controlling for the multiple testing problem. If one took completely random data and then applied this procedure, it would falsely show that some of these random statistics showed a high effect. In fact, since among a random sample of values, by construction, there will always be some that would be "above the average". (Aside: you did not describe how to construct the confidence interval associated to this mean adjusted prevalence score.) Maybe I’m missing something here and there is validity to compute the mean adjusted prevalence score. If so, then please provide some literature to support this method.

Some more general comments about the paper:

6) I don't know what the source of the data is. I am not researcher working in USA, so I am not familiar with this data source.

a) this data source should be reference and either described or a paper which adequately describes this data source should be reference.

b) I don't know where these terms/factors come from. It looks there is a query to the data set program, but I don't know what kinds of queries are being made. I am familiar with ICD codes and know where they come from, but I don't know why or how the authors came up with 18K different factors that were then tested.

7) The type of results reported. I think that if the purpose of this paper is to give information on the factors which contribute to IPV in seniors, then I think more analysis should be done. In the results section of the paper, the authors report that they identified 250 and 1240 terms with high or medium confidence. Then, in tables 1 and 2, they list the top 20 terms. (I did not receive the supplemental file, but I presume the other 1000+ terms are listed there.) My main concern is, “how is a scientist suppose to synthesis this information?” These are a lot of terms here. I would suggest that this information would be more useful if this paper was able to organize or structure this information better. For example, they could cluster the subjects or group together the main terms in some way. (A first pass cluster analysis or factor analysis for example.) I would assume that there is high correlation between these terms. Note that there is a total of only 420 senior IPV subjects. (See Figure 1b) Also, I see that members of this group have done similar work before. (See for example, Hacialiefeniouglu et al, 2021, Scientific Report.)

8) The statistics that are reported in tables 1 and 2. In these tables, the authors report log (base 2) odds ratios and counts. These are not the easiest values to interpret. Usually, one would transform the log odds ratio and confidence interval to the odd ratio scale. The usual population scientist thinks in terms of odds ratios. Most cannot convert from log odds ratios in their heads. (And even the ones that do have a feel for log odds ratios usually works in the natural log scale not in base 2. So, they would have to multiple their usual calibrated LOR by 0.7 to make the conversion.) The raw counts are okay, but the proportions are more convenient for those who want to interpret the results.

6. PLOS authors have the option to publish the peer review history of their article (what does this mean?). If published, this will include your full peer review and any attached files.

Reviewer #1: No

Reviewer #2: **Yes: **Michael Escobar

---

## [Author Response · Author response to Decision Letter 0]

10 May 2022

Please view the attached PDF file to see our responses to the reviewers' comments with better formatting.

Reviewers' comments:

Reviewer #1: Thanks to the authors for taking the time and effort to prepare this article. Here are some comments that the authors can consider for revising their paper.

We would like to thank the reviewer for their constructive feedback. We believe these revisions made the manuscript more readable, informative, and potentially impactful. 

1.1. The mathematical notations used in the article seemed unnecessary in my opinion, an example: the explanation how contingency tables were formed. I suggest authors to find simple analytic language to express their approach concisely and avoid notations when not needed.

- We agree with the reviewer that the elaborate description of methods risks shifting the focus from the application (which is at the focus here) to methodology. However, some of the points raised by Reviewer #2 and the resulting discussion demonstrates that the methodology needs to be carefully considered for our results and the conclusions derived from these results to be reliable. For this reason, we decided to keep the detailed description of the methodology. Please see our response to Reviewer Comment 1.3 for further explanation.

1.2. I am confused why a log odds ratio was used to present prevalence.

We agree with the reviewer that referring to the odds ratios becomes confusing to the readers. To avoid this confusion, we now refer to it as “co-morbidity score”. 

1.3. This does not seem to be a statistical methods paper; I wonder why readily available statistical software were not used to do the analysis, but rather manual calculation was preferred. For example, do the authors think using a software-based analysis would have saved them the space and effort for the section “Accounting for measurement error due to rounding” and made it easier for the readers to focus on the findings more?

While the main aim of the study is not to make a methodological contribution, this work is a result of a collaboration between domain scientists and data scientists, which facilitates the usage of more dedicated methods that are tailored towards the application. Specifically, the statistical analysis done here is tailored towards the estimation of the effect sizes (i.e., the co-morbidities) while accounting for (i) expected variance, (ii) measurement error due to interval censoring, (iii) multiple comparisons, and (iv) selection bias in health records. As such, we are not aware of any statistical software that provides all the functionality that is applied in this work. In addition, as our discussion with Reviewer 2 demonstrates, there are a lot of intricacies in the statistical data analyses. These intricacies are typically obscured by statistical software. Thus, even when a statistical software with proper versioning (and legacy support) is employed, we believe that a detailed mathematical description is essential to keep the results reproducible and to make any potential issues in the analysis apparent. 

1.4. The result section is not elaborate enough. The section mostly refers to tables and does not point out to key and important findings from the work at all.

We added the following results to help interpret the findings: The distributions of co-morbidity and differential co-morbidity scores in Senior and Background populations (Figure 2 in the revised manuscript), the scatter plot of the comorbidity of terms with IPV in the background population vs. the co-morbidity of terms with IPV in the senior population (Figure 3 in the revised manuscript). We added substantial text to Results to help read these figures. In addition, we enriched the Discussion section with more interpretation of the results to address various points raised by both reviewers. To avoid repetition, we did not include interpretation of the results in the Results section.

-The limitation section in discussion should list limitations of the study and not limitations in the topic (those can go in the introduction)

This is a great point, we thank the reviewer for pointing this out. We believe that the difficulty associated with clinician’s identification of IPV constitutes part of the motivation for this study (can we identify markers to help the clinician?), but it is also part of its limitations (most of the IPV cases that go into electronic health records are severe cases, so the markers we find may not be as useful for identifying less severe forms of IPV). To clarify this distinction, we moved the discussion on the limitations of the field to the Introduction and rewrote the Limitations paragraph in Discussion to highlight potential limitations of the sample. We also added the limitations of our findings in terms of providing causal/mechanistic insights into the associations that are identified.

-The conclusion should pinpoint which symptomatology or diagnoses the authors think could be flags for IPV in older women, which is missing. Rather general statements were made.

We manually went through the terms that were identified as differentially co-morbid with IPV in older women as compared to the background population. We annotated and grouped the terms based on their general categories. From each category, we selected terms based on their differential co-morbidity, overall co-morbidity in the older population, and potential clinical relevance. Overall, we report 20 terms in Table 2 of the revised manuscript that can potentially serve as flags for IPV in older women. We added a detailed discussion for the potential clinical relevance of these terms in the Discussion section.

Reviewer #2: First, note that this reviewer is a biostatistician and not a subject matter reviewer. Therefore, most of my review is concerned with the statistical methods used in the analysis. 

We thank the reviewer for their detailed comments and suggestions, which have been very intriguing and helpful in terms of the statistical analysis. Some questions raised by the reviewer prompted us to think about these issues more and develop solutions that can be considered novel from a statistical perspective as well. We provide a long and detailed discussion of our approach below, we will greatly appreciate the reviewer’s insights on the soundness and novelty of our approach.

Overview: This paper looks at the relationship between a large number of possible factors and intimate partner violence (IPV) among seniors. The data come from a large data source and the investigators identify a large number of potential factors that are looked at as possibly being related to IPV. (Initially, there are >18K factors considered.) For each examined factor, the authors basically construct a 2x2 table that is standard in epidemiological research. That is, this table looks at the cases versus the controls which are those who are and are not identified as victims of IPV. For each group (case and control), one counts the number who are exposed and not exposed to the individual factor. After creating a 2x2 table for each factor, they compute the log of the odds ratio and the associated confidence interval. If the confidence interval is greater than 0 (so the odds ratios are bigger than 1), the investigators then basically declare the factor important. Also, the reviewers compare the odds ratio for the senior population to the odds ratio of the non-senior population to see if the effect of each of the factors are discernible higher for the senior population. In the main body of the paper, the authors list some of these top factors. There is mention of a supplementary table with a larger list of the factors which are identifies as important. Note: This reviewer does note that there were some other subtle issues that are addressed in the paper. The above overview is this reviewer’s understanding of the main concepts in the paper.

Overall, I like the general idea of the paper. I think that electronic health records are a largely untapped source of valuable information. However, I have important concerns that this manuscript does not do enough to account for the high number of multiple tests performed for this analysis. In the statistical literature, this is known as the multiple testing problem. Although the authors don’t mention that they are doing formal hypothesis testing, the essence of this problem still applies to this study. It might seem that I am suggesting substantial changes in the analysis; however, they would probably not take long to do.

I would further note, that I did look up some papers that were produced from this general research group. Many of my below comments would have been answered in a satisfactory manner if they used the methods that they used in some other papers. I would specifically point out the paper: Karakurt, Patel, Whiting, Koyuturk, 2017, J Fam Viol 32:79-87, DOI 10.1007/s10896-016-9872-5. I will be referring to this paper as Karakurt et al in the below. 

I will first list some of the statistical notes/comments/issues and then list more general matters.

1) Multiple testing. Using the criteria that a factor is acceptable in some sense (high or medium confidence) if the confidence interval is greater than 0 is basically mathematically equivalent to a null hypothesis test. (Technically, this would be a 1-way test at the .025 level.) So, the authors are doing 18k null hypothesis test (or, considering only the "valid" test, there are 2056 different test). This is the classical multiple testing problem. You are saying that something is important if the test statistic is would be rare if it was random. That is, rare means that the outcome only occurs with about a 5% probability. Then you run this 2000 or so times and you see lots of “rare” things. This is a classic problem and there are many different ways to approach this.

a) In the other paper (Karakurt et al), a Bonferroni correction is made. So, the authors are aware of the problem. When one is doing many test, this method might be too conservative.

b) As an alternative to the Bonferroni, I prefer using the False Discovery Rate procedure (reference: Benjamini and Hochberg 1995 and Benjamini and Yekutieli 1999.) This reviewer is a fan of this method. At the end of the analysis, one gets a set of factors. In this set, it is acknowledge that some might still be by chance but overall the set contains factors which are highly likely to be important. 

We agree with the reviewer that the multiple hypothesis adjustment is an important issue. The reviewer’s comment made us think about this problem and we believe we have found an effective solution for computing multiple hypotheses (FDR) adjusted confidence intervals for a ranked list of terms (as opposed to making a fixed hypothesis test). Our approach is based on the Benjamini and Hochberg (BH) procedure and their more recent work on BH-selected false coverage rate (FCR) adjusted confidence intervals (Benjamini and Yekutieli, 2005). However, to avoid specifying an arbitrary threshold for the hypothesis tests (e.g., a fixed null level OR=1), we have extended the existing FCR adjusted confidence intervals (CIs) to be valid for any null level (essentially to be valid for the selection of top k terms with highest effect size for any selection of k). 

Below, we explain our overall reasoning for this framework, illustrate the issues with the existing approaches and finally provide a detailed explanation regarding our computation of an accompanying interval for the BH-procedure. 

Note that, here, our objective is to rank the terms, uncover the ones with highest co-morbidity and estimate a lower bound on their effect size, as opposed to identifying all terms that pass a certain significance threshold. Thus, our key objectives can be summarized as follows: 

(1) Identification of terms exhibit strongest co-morbidity, and 

(2) Estimation of reliable confidence intervals for the effect size of identified terms. 

Use of Fixed Hypothesis Tests vs. Confidence Intervals

In this work, we wanted to stay away from p-values and statistical significance tests that do not provide interpretable and meaningful estimates of effect size. We rather focused on confidence intervals (CIs), which provide more direct information on effect size (e.g., the co-morbidity measured by odds ratios). Consider the following example that illustrates this point: 

Here, we are comparing the co-morbidity of two terms with IPV in the background population (Term A is something rare: “Injury of tympanic membrane”, and Term B is more common: “Infectious disease”). The null hypothesis for both terms is that the term does not have an odds ratio greater than 1. The significance (shown as z-score since we were not able to plot -log(p) due to the insufficient numerical precision) of the test scores suggests that the null hypothesis is rejected for both terms, and the co-morbidity of term B is more significant than that of term A (first panel). In contrast, if we consider the odds ratios and confidence intervals (second panel), we see that term A has a much greater effect size (i.e., comorbidity) than term B, which is in contrast with the statistical significance comparison taken at face value. 

However, if we look at the frequency of the terms (last panel), we see that the reason statistical significance gives misleading information on the comparison is because term B is much more common than term A, which allows term B to be deemed more significant than term A despite exhibiting a much lower effect size. Thus, we conclude that while statistical significance can let us deduce that both terms A and B have odds ratios significantly greater than 1, it is not very informative about their effect sizes and is not appropriate for making a comparison. But, this leads us to the question: What can we conclude about their effect sizes and how can we make a meaningful comparison between the terms?

The above example demonstrates what we wanted to take into account while ranking the terms for their “interestingness” (basically we aimed to uncover the terms with high effect size, rather than common terms with low effect size). For this purpose, we wanted to prioritize the effect sizes after disregarding their unreliable portions (i.e., their expected deviation/error). That’s why, we ranked the terms according to the minimum bound of their confidence intervals (rather than p-values or statistical significance) and analyzed the top terms that are uncovered.

Multiple comparisons problem while estimating confidence intervals and existing work on FCR adjustment

Regardless of the issue discussed above, we agree with the reviewer that, in essence, multiple comparison problem still applies: After such a sorting is applied and top terms are taken, the confidence intervals are no longer valid (not valid in the sense that the lower bound of the interval no longer implies statistical significance, neither FWER or FDR is bounded) and an adjustment is required. 

To resolve this problem, we considered a more recent work (Benjamini and Yekutieli 2005) that aims to produce confidence intervals adjusted for false coverage rate (FCR) for the items identified as significant by the Benjamini-Hochberg (BH) procedure. The core idea seems simple: If after applying BH procedure on M items, R number of tests are rejected, constructing 1 - alpha * R/M level intervals will bound the FCR of the R items (equivalent to bounding the FDR with BH procedure) by alpha. 

Limitations of existing approaches for FCR adjusted CIs

After applying the Benjamini and Yekutelli’s procedure to our dataset, however, it became clear that this FCR controlling procedure is not sufficient for our purposes: Among the 5464 valid terms (with non-zero frequency), a large number (4664 terms) are deemed significant by the BH procedure (comparing with OR=1, for alpha=0.05). When we inspected the results closely, we observed that the application of the BH procedure results in a minimal adjustment in the confidence intervals (e.g., FCR adjusted interval for the odds ratio of “Injury of tympanic membrane” is [37.0, 66.3], while the unadjusted confidence is [37.5, 65.3]). Thus, it does little to resolve the multiple comparison issue when the top terms are considered. This is because the FCR adjustment procedure assumes we will equally consider all identified terms while this is certainly not practical. 

A way to alleviate this issue, we thought, is to make more stringent tests e.g., instead of making a test for the null hypothesis OR>1, making tests for OR>3, OR>5 or OR>10 and marking the identified/significant terms as Minor/Moderate/Highly associated with IPV. For context, these more stringent levels are marked in the histogram of co-morbidity scores across all terms below:

Here, we observe that the histogram is shifted to the right (not centered around OR=1) and OR>3 seems like a more natural null level rather than OR>1 (this is likely because of a selection bias in the dataset, please see below our answer for point #5 for a more detailed explanation on why we think OR=3 is a more appropriate null level indicating no association). Here, OR>5 and OR>10 represent arbitrarily selected, more stringent levels approximately 0.75 and 1.75 standard deviations above the mean. 

After applying the BH procedure with these more stringent “null levels”, 1487/582/162 terms were deemed significant for respectively OR=3/5/10, (e.g., the corresponding FCR adjusted intervals for “Injury of tympanic membrane” are respectively: [34.8, 70.5], [33.1, 74.0], [31.3, 78.2]). Notice that, as the stringency of the test level increases, FCR adjusted intervals become wider and the lower bound of the interval keeps decreasing, thus, these intervals are still not valid in determining the statistical significance. That is, we can conclude that “Injury of tympanic membrane” has odds ratio higher than 10, but cannot conclude that it has odds ratio higher than 31.3. Thus, with this approach, we cannot know where exactly the minimum level that we can reasonably conclude lies between 10 and 31.3. Plus, the 3/5/10 levels are still somewhat arbitrary, a term with OR=9.99 still fails the test while one with 10.0 passes, which is another inherent limitation of this approach (though it is worth noting that our aim is not to find all significant terms, but rather to identify the most significant/notable ones and to estimate a bound on their effect sizes). 

In the example of “Injury of tympanic membrane”, to find the maximum level Q where we would still identify this term as significant after BH-procedure for OR>Q, we can apply a binary search between 10 and 31.3. This analysis suggests that maximum such Q is 27.9 and this is where BH-procedure and lower bound of the FCR adjusted interval starts to agree (adjusted interval: [27.9, 88.0]). Thus, this is the interval that we consider to be accurate (lower bound indicating a level of significance that is appropriately FDR controlled) for a ranked list of terms in descending order (according to the lower bound of confidence interval, or equivalently maximum Q). According to this ordering, it turns out “Injury of tympanic membrane” is 7th most significant term (where 6 of the terms can be deemed significant for Q>27.9 level tests, and the others are not significant at 27.9 level). 

In light of these observations, we conclude that the BH-selected FCR adjusted confidence intervals have a critical limitation: They only give a guarantee about the average of the findings (average false coverage, or false discovery), but does nothing about the individual errors. What’s more concerning is that these errors are not uniform, but seem to be concentrated around the ones exhibiting the highest effect size. 

For example, if we test for OR>1 and generate FCR adjusted CIs, there are 4664 significant terms and the intervals are narrow ([37.0, 66.3] for “Injury of tympanic membrane”). But, if we test more stringently for OR>10, there are much less significant terms (162 terms) and the intervals are suddenly much wider ([31.3, 78.2]). What seems to be happening here is that, in the former case, the terms exhibiting low effect size are amortizing the errors of the ones exhibiting higher effect size, and in the latter case, this amortization is much lower. In both cases, however, the resultant confidence intervals are unreliable because we cannot reliably use them to make conclusions about the ones having high effect size (e.g., we cannot conclude “Injury of tympanic membrane” has odds ratio > 31.3 despite that’s what the interval is suggesting to us). In our opinion, this is most troubling because, while reviewing the results, our attention will not be random: We will inevitably focus on the ones with the highest effect size — because that is the point of the research in the first place!

To overcome these limitations, we decided to employ a simple approach: 

We consider all possible null levels (denoted Q), construct BH-selected FCR adjusted intervals for each level (for significant terms), select the most stringent interval for each term (this is necessarily at the highest level Q a term would be deemed significant), sort the terms according to the minimum bound of their intervals, and make conclusions about their co-morbidity based on this. 

Please see below for the details of this procedure. 

Our computation of accompanying CIs that are adjusted for FDR

Overall, we generalize the 3/5/10 stringent testing approach (described above) and test for all possible thresholds Q with logarithmic increments of 0.01 to find the maximum Q(t) and the corresponding FCR adjusted intervals which are controlled for FDR for a given term t. This carries the interpretation that the lower bound of the confidence interval will indicate the maximum level of odds ratio that would be deemed significant after adjusting for FDR using BH procedure. Since these intervals are necessarily wider than or equal to the FCR-adjusted BH-selected CIs (for any given null level), they follow the properties detailed in Benjamini and Yekutieli, 2005 and are controlled for FCR. The main benefit of this approach is that it avoids the need for specifying a null level and allows it to be determined a posteriori at the expense of some statistical power (since all possible null levels are considered simultaneously). 

Below, we also formulate this as a computational problem as follows: 

For each term t, find the maximum Q(t) such that OR(t, IPV|Z) is found to be significantly larger than Q(t) at alpha (e.g., 0.05) level after BH-procedure is applied to adjust for FDR. 

Sort each term in descending order according to Q(t) and get its ranking R(t). 

Construct FCR adjusted confidence interval for the odds ratio of each term at alpha * R(t) / M level where M is the total number of valid terms. The lower bound of this interval will be equal to Q(t). Since the distribution of log-odds ratio is symmetric, the upper bound of this interval can also be found from 2.^(2 * LOR(t, IPV|Z) - log2(Q(t))).

While it may be possible to compute step 1 more efficiently through other means, here we simply apply all thresholds Q in a brute force manner to find Q(t) and to construct the adjusted intervals. 

Implications of this procedure and differences from BH-selected FCR adjusted CIs

BH-selected FCR adjusted confidence intervals (Benjamini and Yekutieli, 2005) control the average false coverage error when K number of terms are selected via the BH procedure (e.g., if K = 100, the expected number of total false coverage errors would be less than 5 for alpha=0.05). Yet, they provide no guarantees as to how those errors would be distributed (where are those 5 errors? Are they distributed 0.05 for each 100 terms, or is it expected that almost certainly those errors would be on the top 5 most significant terms?). The extended procedure aims to address this issue. 

We reason that by the repeated application of all possible null levels (Q), we are considering every possible threshold level between terms (that would make one significant, and the other not, unless they are both equivalently significant) and adjust the FCR for each of those levels. Thus, we reason that this would bound the FCR of the first k terms for every k. For example, the selection of the first term (denoted t1) would guarantee the FCR(t1)<=0.05, thus, the total expected error to be less than 0.05. The selection of first two terms, would bound the total expected error of the first two terms to be less than 0.10 and so on. This is the main guarantee these extended intervals provide. Thus, this allows the selection of k (or the significance level Q) to be done posteriori, eliminating the need for specifying a null hypothesis and allows more direct conclusions to be made about the effect size (e.g., by looking at the minimum bound of the confidence interval, we can now determine whether a term would be deemed significant at any specified level, thus, eliminating arbitrary thresholds). 

Moreover, we reason that there is an additional guarantee that these intervals might be providing (intuitively): If the errors are expected to be in decreasing order (the first term having the highest expected error and so on), then we can guarantee that the expected false coverage error of each individual term would be bounded by 0.05, thus achieving a uniformity condition (uniformity in the sense that the confidence interval of each term would provide the same minimum coverage guarantee e.g., 95%). Note that, we believe that it is reasonable to expect that the errors would be in decreasing order because the first term corresponds to the top term exhibiting highest association (therefore, if the null hypothesis were to be true for all terms, the top term would be the one exhibiting the highest deviation, thus, having the largest error). Please see below for an illustration of this intuitive explanation. 

2) The issue that the observations are only know to the nearest "tenth". Here, the authors use a very conservative correction. When creating the log odds ratio (LOR) and the standard error estimate, they assume that the actual value is the worse possible. That is, the counts are rounded off to the nearest 10.

a) In the other paper (Karakurt et al), the authors use a Monte Carlo simulation method to sample the possible values which would be the observed value plus or minus 5. That seems very reasonable to me. Basically, this approach is a type of multiple imputation used for missing data. (Also note, the authors in that paper then compute an adjusted confidence interval. This adjusted confidence interval can be used to approximate an adjusted standard error for the LOR.)

b) In the other paper (Karakurt et al), take the sampled values are used to create an estimated confidence interval. Instead of that, one option is that one can use multiple imputation methods. That is, take the sampled values and then use multiple imputation methods to get an estimate of the standard error of the LOR which accounts for the interval censoring of the data. (see for example equations 14.7-14.10 in this article: https://www.sciencedirect.com/topics/mathematics/multiple-imputation for one reference for these formulas. These formulas are available many other places.)

c) Instead of random sampling, one can do some calculations based on the fact that one is sampling from finite discrete distribution. That is, if one is planning on sampling an integer between -5 and 5, then just calculate the LOR and standard error for each of these 11 points and then use the formulas in (b) above.

Based on the reviewer’s suggestion (option b), we sampled values and used multiple imputation methods to account for the uncertainty due to interval censoring of the data. Since this results in a less conservative method for interval censoring than we previously applied, and since now we additionally account for multiple hypothesis testing, the overall effect on the results was minimal, but now the statistical framework is much improved. We thank the reviewer for this suggestion.

3) Prevalence: The term prevalence has a well-defined definition in the epidemiology and population health sciences. It is the number of events in a population at a designated time. There is a similar definition of the prevalence rate. The use of the term "prevalence score" to mean the log of the odds ratio is very confusing and is a misuse of well-established terms.

This is an issue brought up by both reviewers and we agree. We renamed the prevalence score as the co-morbidity score. 

4) Difference between the log odds ratios between the senior population and the younger population. On page 4, the authors look at “Assessment of Differential Prevalence”. They define this term in equation 5 and then give their formula for the confidence intervals in equation 6. This is a very reasonable thing to do. It is the difference between the two LOR. It is the increase in the odds between the senior population versus the non-senior population. This is a very common statistics to look at therefore the statistical properties are well known. The authors need to change the way that they compute these confidence intervals.

Note the following statistical properties: a) the LOR statistics are approximately normally distributed with the standard errors as given in the equation 2. b) Since the senior population and the younger population are completely different populations, then these samples are statistically independence. c) Therefore, the difference of the two statistic defined in equation 5 has an approximate a normal distribution. d) standard error of difference of between the sampled log odd ratio's between these two populations is just "square root of the sum of the squared standard errors". That is, if the standard errors are S1 and S2, then the standard error of the differences is sqrt( S1^2 + S2^2). One can then get the confidence interval of the difference in the log odds ratio by the methods already discussed in the paper. (That is, it is the difference +/- 1.96* standard error of the difference as described in equation 3).

Based on the reviewer’s suggestion, we have updated the formula to compute the standard error of the differential co-morbidity (previously named differential prevalence). 

5) I don't see what the value of equation 4. That is, the value of using a mean adjusted prevalence score which is obtained by subtracting the LOR by the average of the LOR and then using that to declare "high" effect. That seems like an attempt to control for the multiple testing, but does not appear to have any validity in controlling for the multiple testing problem. If one took completely random data and then applied this procedure, it would falsely show that some of these random statistics showed a high effect. In fact, since among a random sample of values, by construction, there will always be some that would be "above the average". (Aside: you did not describe how to construct the confidence interval associated to this mean adjusted prevalence score.) Maybe I’m missing something here and there is validity to compute the mean adjusted prevalence score. If so, then please provide some literature to support this method. 

The purpose of equation 4 and mean adjusted scores is not to attempt to control for multiple testing problem (though we agree they were somewhat helpful in that regard before we added the FDR adjustment), but rather, it is to alleviate the effect of possible selection bias in this database. Please see below for our reasoning (in the next page):

Here, the left panel shows the histogram of the term associations (i.e., odds ratios) with IPV in the Senior population, and the right panel the same histogram in BG population. As it can be seen, the histograms are considerably shifted to the right, such that the mean & median odds ratio are around 3 in both Senior and BG populations. We had observed the same phenomenon in our previous work as well (again using the IBM Explorys database) [1] and had investigated why this might be so (for this purpose we had investigated a bunch of control conditions in addition to IPV and had repeated the analysis for each condition). Our observation was that this overrepresentation (i.e., shift to the right) was there for each condition and the magnitude of the shift (i.e., mean LOR) was typically higher for rarer conditions (Figure 2 in [1]). 

Overall, we believe this happens due to two reasons:

Some of the terms that are negatively associated are not observed (i.e., there are no records having both IPV and that term), thus, are excluded from the analysis. This exclusion causes a shift in the mean association (i.e., since some negatively associated terms are censored, the mean association becomes positive). 

There is a selection bias in the record keeping, particularly for more severe or rare conditions. If a severe condition is detected, the record keeping is done more meticulously or simply more medical tests are performed and more terms (i.e., medical conditions) are detected. Otherwise, there is less scrutiny and many terms fly under the radar. This may explain why we observe a positive association between almost all terms. The issue, in this case, is that the association that we identify (or at least some portion of it) is not necessarily due to an inherent comorbidity in the population, but rather, due to reasons related to term detection or record keeping. Thus, a correction is required. 

We had investigated whether (1), the statistical explanation, can account for the big portion of the observed shift. Our results (Figure 3 in [1]) had suggested that while negatively associated terms are indeed censored (particularly for rare terms), this can only account for a relatively small portion of the shift. Thus, we had concluded that there is likely some selection bias and an adjustment is required. We had performed this adjustment by measuring the mean LOR of all terms and subtracting it from the LOR of the terms (such that the LOR distribution would be centered around 0). 

In a way, what this adjustment achieved was: (i) Instead of considering the null level of no association to be OR=1, consider it to be the mean LOR of all terms (~OR=3), and (ii) While estimating the effect size (i.e., the co-morbidity), only consider the odds ratio beyond the null level (i.e., OR=3). For example, if the confidence interval for the OR of a term X is [6, 20], do not conclude the co-morbidity to be higher than 6, but instead conclude the co-morbidity of X to be higher than 6/3 = 2 (i.e., expect the term to be at least 2 times more prevalent in victims of IPV). 

Note that, we acknowledge that this can be a rather conservative approach for estimating the effect size, that’s why, we reported both the adjusted and raw (unadjusted) co-morbidity scores. In the revised version, we now make some modifications and additions to this approach. First, to make the adjustment more easily understandable to the readers, we now make the adjustment according to OR=3 level (which we consider to be an appropriate null level) rather than exactly the mean odds ratio (which is around OR=3.2). In addition, to make the mental processing of the results easier, we now assign some simple labels to the terms according to their association levels: That is, we consider the terms with OR significantly higher than 3/5/10 to be Minor/Moderate/Highly comorbid terms (corresponding to at least 1/1.67/3.33 adjusted co-morbidity).

Overall, we believe that these changes (along with the FDR adjustment and the changes in the calculations of the interval censoring adjustment and differential co-morbidity) have markedly improved the statistical framework as well as the interpretability of the results. We are thankful to the reviewer for their constructive comments and suggestions. 

[1] Yılmaz, Serhan, et al. "Identifying health correlates of intimate partner violence against pregnant women." Health information science and systems 8.1 (2020): 1-13.

Some more general comments about the paper:

6) I don't know what the source of the data is. I am not researcher working in USA, so I am not familiar with this data source.

a) this data source should be reference and either described or a paper which adequately describes this data source should be reference.

We added the following text and associated references to the Introduction to clarify this point:

“To answer these questions, we utilize electronic health records (EHRs) provided by the IBM Explorys Therapeutic Dataset [1]. IBM Explorys is a private Electronic Health Record (EHR) database, which pools data from more than 8 billion ambulatory visits to more than 40 US healthcare networks including diverse institutions and points of care [2]. It is a browser-based search engine with query options of various diagnostic categories based on ICD-9/10 codes. Cohorts include data on diagnoses, findings, and demographics. In this paper, we use diagnostic data we obtain by querying this tool. Throughout this paper, we refer to diagnoses, findings, and demographics returned by Explorys as "terms".”

[1] IBM Explorys Theurapetic Dataset. https://www.ibm.com/downloads/cas/NNPN9J9Q

[2] IBM Explorys EHR Solutions. https://www.ibm.com/products/explorys-ehr-data-analysis-tools

b) I don't know where these terms/factors come from. It looks there is a query to the data set program, but I don't know what kinds of queries are being made. I am familiar with ICD codes and know where they come from, but I don't know why or how the authors came up with 18K different factors that were then tested.

The terms are not a query to the database system, but rather, provided as a result of the query. For example, here is the process: (1) We specify the conditions of a population in the query (e.g., women, has IPV, ages 18-65), then (2) the system returns a list of terms (~18k terms) and their corresponding number of records in the specified population for each term. In total, we make 4 queries with the following conditions: 

BG query: Women, ages 18-65

IPV in BG query: Women, has IPV, ages 18-65

Senior query: Women, ages 65+

IPV in Senior query: Women, has IPV, ages 65+

In the methods section, we now clarify that a list of terms T (along with their frequencies) is returned with the query. 

7) The type of results reported. I think that if the purpose of this paper is to give information on the factors which contribute to IPV in seniors, then I think more analysis should be done. In the results section of the paper, the authors report that they identified 250 and 1240 terms with high or medium confidence. Then, in tables 1 and 2, they list the top 20 terms. (I did not receive the supplemental file, but I presume the other 1000+ terms are listed there.) My main concern is, “how is a scientist suppose to synthesis this information?” These are a lot of terms here. I would suggest that this information would be more useful if this paper was able to organize or structure this information better. For example, they could cluster the subjects or group together the main terms in some way. (A first pass cluster analysis or factor analysis for example.) I would assume that there is high correlation between these terms. Note that there is a total of only 420 senior IPV subjects. (See Figure 1b) Also, I see that members of this group have done similar work before. (See for example, Hacialiefeniouglu et al, 2021, Scientific Report.)

We thank the reviewer for their insightful suggestion that also takes into account our other efforts in this domain. Unfortunately, unlike the data utilized in the Scientific Reports paper, the EHR data from Explorys does not provide specificity at the level of individual records (we only get information on the number of records for the terms and their overlap with IPV, not individual records, and we also do not have information about the number of records that overlap between the 18k terms). Thus, we are not able to cluster records based on their symptomatology, which we agree would provide additional insights that could help interpret some of our findings. We note this as a limitation of the current study in the Discussion section of the revised manuscript.

As for synthesizing the findings to provide more structured information to the readers, we have manually gone through the terms to identify potential flags for IPV in older women (these are reported in the revised Table 2). We have also annotated and grouped them based on their general categories to avoid duplicate reporting of highly similar terms. Please see our response to the last comment by Reviewer 1 for details. 

8) The statistics that are reported in tables 1 and 2. In these tables, the authors report log (base 2) odds ratios and counts. These are not the easiest values to interpret. Usually, one would transform the log odds ratio and confidence interval to the odd ratio scale. The usual population scientist thinks in terms of odds ratios. Most cannot convert from log odds ratios in their heads. (And even the ones that do have a feel for log odds ratios usually works in the natural log scale not in base 2. So, they would have to multiple their usual calibrated LOR by 0.7 to make the conversion.) The raw counts are okay, but the proportions are more convenient for those who want to interpret the results.

We agree with the reviewer and thank them for pointing out this important point. We now use odds ratios instead of LORs throughout the paper.

---

## [Editor Report · Decision Letter 1]

18 Jul 2022

PONE-D-21-21436R1Adverse health correlates of intimate partner violence against older women: Mining electronic health records

PLOS ONE

Dear Dr. Karakurt,

Thank you for submitting your manuscript to PLOS ONE. After careful consideration, we feel that it has merit but does not fully meet PLOS ONE’s publication criteria as it currently stands. Therefore, we invite you to submit a revised version of the manuscript that addresses the points raised during the review process.

The authors have responded to the concerns expressed in the previous round of review comments in the revised manuscript. The revised manuscript is somewhat improved—thank you. However, the authors should address the following issues further before this manuscript is considered for publication.

First, the study results are interesting. However, there is significant room for improvement regarding the appropriateness of the scientific method in the publication criteria of PLOS ONE.I agree with comment 1 of the previous Rev #1. In fact, many formulas can be omitted. For example, the formulas for calculating crude odds ratios and 95% confidence intervals from a 2 x 2 contingency table can be found in regular textbooks (e.g., equations 1, 2, 3, etc.), so there would be no need to include them in the text. Since PLOS One is a journal with readers from various educational backgrounds, please explain in writing what you have done unless it is absolutely necessary to use mathematical formulas. Since this paper is not a paper on statistical methods, writing formulas would rather spoil the merit of this paper. This applies to other parts of the method as well. If the authors want to write the formulas described in Revision 1, upload them as Supplemental Material.As mentioned in response to Rev #1, there are differences in the results from different statistical software, or discrepancies in the calculation results due to different versions of the software, even within the same software. Therefore, the code or program used by the authors (in C or any other language, e.g. R code) itself should be uploaded as Supplementary Material.The censoring that the authors refer to would be better described as rounding or rounding off. Censoring, as usually used in cohort studies, usually refers to a participant dropout or exceeding the upper measurement limit for counts or measurements. Please reconsider your use of the term. Also, please remind us in the methods section that rounding occurs in the dataset.There needs to be a more detailed written explanation of how the Monte Carlo simulations were performed. Also, to what extent would incorporating this procedure change the results in the first place? Sometimes it is more accurate to dare to do nothing. At a minimum, the authors need to cite papers that use similar methods and claim their effectiveness. They should also show the results of bypassing this procedure as a sensitivity analysis to demonstrate the effectiveness of this procedure. If there is no significant change (i.e., the number of words extracted is zero or doubled), then it would be simpler and better to omit this process. For example, suppose the height is 171.8 cm. Here, the true value would be between 171.75 and 171.85 cm, but one would not normally use the Monte Carlo method in analyzing such data; as the number n increases, one would expect the overall rounding error to average out (law of large numbers).Is the formula in lines 162~163 the Rubin formula used to merge multiple completion data sets into a single value? As described, the formula itself does not need to be described, but a citation showing how it was conducted is required.Line 182: The term "adjust" usually means adjusted for covariates in cohort studies. Therefore, "adjust" is confusing usage here and a more appropriate term should be used consistently. The consistent use of terminology applies to other parts of this paper as well.Line 199: Although there is a citation above, it is confusing because it suddenly appears to apply here with little explanation of the BH-procedure. What the reader needs is a general explanation of how the BH-procedure can be of benefit when applied in a case like this one. For example, what characteristics does it have compared to the Bonferroni-correction or compared to the Holms-method? (Please emphasize that the BH method allows for some familywise error rate, but suppresses beta error. ) Also, Bonferroni corrects the overall type I error by using a new value of α divided by the number of repetitions of the significance-level α as a new value of α. As already mentioned, the policy of explaining in writing rather than in equations applies to the other parts of the method as well.The null hypothesis level is placed as OR=Q, but it is complicated whether it is q or Q, in addition to not defining FDR=q used in the BH-method. The authors seem to express it as Q, which later appears as μ, but the null hypothesis is only OR=1, which is less confusing. If the authors will perform this procedure, I want to see it performed with q properly defined as the null hypothesis with OR=1 after correcting for the selection bias that the authors refer to.Line 243, I am confused because μ=3 suddenly appears. The authors should write a reason why μ=3 can be placed. If they say it is determined by geometric mean (line 287), that should be written in the method. Also, the author suddenly decides OR=3 very roughly here (line 287: approximately centered around OR=3), while the other parts are calculated using complicated methods, which is very unbalanced and confusing from the rigor of the other parts.The authors use the term selection bias at μ=3. Is this a selection bias in the first place? Usually in cohort studies, selection bias refers to a bias that arises at the sampling stage, such as targeting a certain biased population. in line 228: (e.g., if a severe condition is detected, more medical tests may be performed which can lead to the detection of more terms.) is not a selection bias but more an intrinsic aspect related to the use of terms. Also, if the phenomenon is caused by the mechanism as described by the authors, then the higher OR should be corrected to a greater extent and the lower OR to a lesser extent, and a method of correction such as uniformly dividing the OR would not seem to be appropriate.Result section: Some results include matters that should be written in the Methods section, and some include subjectivity that should be discussed in the Discussion section. For example, the procedure of dividing the OR into three levels by looking at the distribution should be indicated to the reader in advance in the method because of the procedure.Line 295: The figure (Fig. 2) cut by OR=3/5/10 could have been drawn by simply calculating and plotting the crude OR for each word. In this case, the OR categories would be multiplied by 3 to 9/15/30.... Similarly, the top 20 could have been calculated by omitting the procedure in lines 186~234. Why not just cut out the process in the middle except for the correction for multiple testing? As it stands, the correction for μ is also a parallel shift, and the effectiveness of the Monte Carlo method is still unclear. I prefer to reanalyze the data based on Occam's razor and the principle of scientific frugality. This way, the conclusion would not be so different.Fig 3 & Table 1: It says "Adjusted for FDR", but isn't what is adjusted for what the authors call selection bias? Adjusting for multiple testing does not change the OR itself, only the confidence interval. Throughout, the use of the terminology should be organized.Figures should exist independently as figures, and it is preferable to redefine the abbreviations used in the text with captions, etc.Please submit your revised manuscript by Sep 01 2022 11:59PM. If you will need more time than this to complete your revisions, please reply to this message or contact the journal office at plosone@plos.org. Please include the following items when submitting your revised manuscript:A rebuttal letter that responds to each point raised by the academic editor and reviewer(s). You should upload this letter as a separate file labeled 'Response to Reviewers'.A marked-up copy of your manuscript that highlights changes made to the original version. You should upload this as a separate file labeled 'Revised Manuscript with Track Changes'.An unmarked version of your revised paper without tracked changes. You should upload this as a separate file labeled 'Manuscript'.

We look forward to receiving your revised manuscript.

Kind regards,

Kenta Matsumura

Academic Editor

PLOS ONE
---

## [Author Response · Author response to Decision Letter 1]

3 Nov 2022

Please see the attached pdf file for our response to the reviewers.

---

## [Decision Letter · Decision Letter 2]

3 Feb 2023

Adverse health correlates of intimate partner violence against older women: Mining electronic health records

PONE-D-21-21436R2

Dear Dr. Karakurt,

We’re pleased to inform you that your manuscript has been judged scientifically suitable for publication and will be formally accepted for publication once it meets all outstanding technical requirements.

Kind regards,

Habil Otanga, Ph.D

Academic Editor

PLOS ONE

Additional Editor Comments (optional):

Reviewers' comments:

Reviewer's Responses to Questions

**Comments to the Author**

1. If the authors have adequately addressed your comments raised in a previous round of review and you feel that this manuscript is now acceptable for publication, you may indicate that here to bypass the “Comments to the Author” section, enter your conflict of interest statement in the “Confidential to Editor” section, and submit your "Accept" recommendation.

Reviewer #1: All comments have been addressed

Reviewer #2: (No Response)

2. Is the manuscript technically sound, and do the data support the conclusions?

Reviewer #1: Yes

Reviewer #2: Yes

3. Has the statistical analysis been performed appropriately and rigorously? 

Reviewer #1: I Don't Know

Reviewer #2: Yes

4. Have the authors made all data underlying the findings in their manuscript fully available?

Reviewer #1: No

Reviewer #2: Yes

5. Is the manuscript presented in an intelligible fashion and written in standard English?

Reviewer #1: Yes

Reviewer #2: Yes

6. Review Comments to the Author

Reviewer #1: The authors seem to have addressed all reviewer comments. No major comments from my part. However, I would have specified "health conditions" instead of just "conditions" in lines 54 and 56.

In line 185, authors mention of "detection bias". I would have termed it as "misclassification bias": a term commonly used in epidemiology for this bias category. I agree with the previous reviewer that the use of statistical formulas seem unnecessary at some parts, but authors have made valid arguments to justify their inclusion of formulas.

Reviewer #2: I was the 2nd referee for the first version of this paper.

I think this paper covers an important issues and collected some important data. As for the analysis, they looped through many important possibly important health correlates of IPV. They controlled for the multiple testing problem and the fact that the data was rounded to the nearest tens. In the body of the main paper they listed the strongest correlates and included the data for the other correlates/ variables that they collected.

There are some parts of the analysis that I still find odd. This is the third revision of the paper and some of these issues have been in the first draft and other referees besides me have pointed these outs. There seems to be a stand still on these points. I was tempted to vote that the manuscript be rejected. However, as I pointed out above, this is an important paper and I think the overall message is important.

Here are some of the details that I have issues with:

1) The authors insist on including the definition of the log odds ratio and is variance in the main body of the paper. They said that it is important since some might not know about this. Please note that this paper is clearly in the area of trauma epidemiology. I have been in the field of epidemiology and biostatistics for several decades, and this material was standard when I started graduate school. With all due respect, this seems rather odd in a major journal like this one.

2) Detection bias. I am not sure what to make of this. Large administrative data has variables which are under reported. It does seem that detection bias is another variant of this. However, it is not clear how to adjust for it. I think that one would simple note this as a limitation of using administrative/ eHR data. (I see that they have a prescription, and that is my next point.)

3) It seems that they have decided to select only those correlates where the odds ratio is statistically significant to the average odds ratio of all the correlates. (Note: here the average is the geometric mean. Also, rejecting the null hypothesis at the 5% level is mathematically equivalent to having the average not contained in the 95% confidence interval.) I don’t understand what the rational is for doing this. I don’t think that the authors are suggesting that this should be a standard methodological practice.

4) In item 3 above, the criteria turns out to be to consider terms where the odds ratio is significantly bigger than 3. Please note that in general an odds ratio is a rather odds ratio. For example in some studies, the odds ratio between high blood pressure and stroke is around 2 or 3. So, in many studies, this would be a big effect. By only looking at very large effects, you risk only finding effects which are so big that most people in the field are already aware of these risk factors. Sometimes, the real value of these studies is finding small signals that people were not aware of.

I would be more concerned about items 2-4, except the authors do include some excellent supplemental materials which give the values for these smaller signals. Also, it is the case that with such studies that contain a lot, it is the authors prerogative to concentrate on a part of the results in the body of the paper. I would feel better about this discuss if they simply said that they were going to discuss the biggest effects and interested readers could look at the supplemental material for other details.

7. PLOS authors have the option to publish the peer review history of their article (what does this mean?). If published, this will include your full peer review and any attached files.

Reviewer #1: **Yes: **Riffat Ara Shawon

Reviewer #2: No

---

## [Editor Report · Acceptance letter]

14 Feb 2023

PONE-D-21-21436R2 

Adverse health correlates of intimate partner violence against older women: Mining electronic health records 

Dear Dr. Karakurt:

I'm pleased to inform you that your manuscript has been deemed suitable for publication in PLOS ONE. Congratulations! Your manuscript is now with our production department. 

Kind regards, 

on behalf of

Dr. Habil Otanga 

Academic Editor

PLOS ONE